# Pattern-enhanced Resonant Soft X-ray Scattering for *Operando* monitoring of electrochemical solid-liquid interfaces

Haoyi Li [1,8,9], Kas Andrle [2,9], Qi Zhang [2,3,9], Isvar A. Cordova [3,4,9], Yao Yang [5], Zhengxing Peng [2], Feipeng Yang [4], Guillaume Freychet [4], Scott Dhuey [6], Alexander Hexemer [4], Brett A. Helms [2,6], Weilun Chao [2,3], Bruno La Fontaine [2,3], Ricardo Ruiz [2,6], Jinghua Guo [4], Wanli Yang [4], Junko Yano [1,7] & Cheng Wang [2,4] ✉

Unveiling interfaces at sub-nanometer scales is essential for advancing the understanding of complex chemical transformations. However, characterizing solid-liquid interfaces with high dimensional sensitivity and temporal resolution remains challenging, due to their dynamic nature and inaccessibility by conventional probes. Here we present an approach, Pattern-enhanced Resonant Soft X-ray Scattering, to overcome the challenges. Rooted in a "sample-as-optics" philosophy, this technique utilizes precisely engineered line-grating nanopatterns to modulate near-field X-ray illumination, coherently enhancing scattering signals from the line-gratings. We implement the method using Ni line-grating nanopatterns in electrochemical water oxidation. The periodic nanostructures serve as diffractive optical elements to reveal the Ni oxidation gradients and structural dynamics at the electrode-electrolyte interfaces. Finite-element simulations corroborate the observed trends by modeling variations in compositions and structures during electrocatalysis. Through integrating advanced sample design with coherent wave nature of soft X-rays, our approach opens accessible pathways to *operando* exploring chemical evolution and sub-nanometer dimensional variations simultaneously in electrochemical systems. This non-destructive method is efficient and element-specific, making it valuable for probing chemical and dimensional dynamics with appropriate modeling.

Great attention has been drawn to a comprehensive understanding of solid-liquid interfaces at atomic scales in a wide range of chemical and biological systems[1–4]. Although modern material characterizations have seen significant advances in resolving interface and interphase properties[5–7], they are largely based on improvements in probe sources and detectors[6,8–10]. In the meantime, devices are becoming smaller and faster, and interfacial structures are becoming more complex and deeply buried[11,12]. However, an effective characterization technique for

[1]Liquid Sunlight Alliance, Lawrence Berkeley National Laboratory, Berkeley, CA, USA. [2]Materials Sciences Division, Lawrence Berkeley National Laboratory, Berkeley, CA, USA. [3]Center for X-ray Optics, Lawrence Berkeley National Laboratory, Berkeley, CA, USA. [4]Advanced Light Source, Lawrence Berkeley National Laboratory, Berkeley, CA, USA. [5]Department of Chemistry and Chemical Biology, Cornell University, Ithaca, NY, USA. [6]Molecular Foundry, Lawrence Berkeley National Laboratory, Berkeley, CA, USA. [7]Molecular Biophysics and Integrated Bioimaging Division, Lawrence Berkeley National Laboratory, Berkeley, CA, USA. [8]Present address: Stanford Synchrotron Radiation Lightsource, SLAC National Accelerator Laboratory, Menlo Park, CA, USA. [9]These authors contributed equally: Haoyi Li, Kas Andrle, Qi Zhang, Isvar A. Cordova. ✉e-mail: cwang2@lbl.gov

studying interfaces with high dimensional sensitivity and temporal resolution under realistic conditions remains a formidable challenge. For example, *operando* transmission electron microscopy (TEM) enables real-time visualization of structural transformations[13,14], but prolonged exposure to high-energy electrons could cause severe damage to the sample, potentially altering the structures and chemistry of interests[15,16]. Moreover, TEM often captures localized regions that may not be statistically representative of the entire interface[17,18]. In contrast, synchrotron-based surface-sensitive techniques, such as modulation excitation X-ray absorption spectroscopy (XAS)[19], ambient-pressure X-ray photoelectron spectroscopy[20,21], and X-ray fluorescence microscopy[22,23], offer robust averaged chemical and structural information. Nevertheless, these methods often difficultly isolate and monitor reaction fronts, particularly at the (sub-)nanometer dimension and (sub-)millisecond timescales[24–26]. These limitations highlight the urgent need for a non-destructive *operando* characterization technique capable of investigating solid-liquid interfaces with remarkable dimensional sensitivity, temporal resolution, statistical reliability, and element-specific sensitivity.

Here, we present a methodology, Pattern-enhanced Resonant Soft X-ray Scattering (PE-RSoXS), for achieving significant spatial and chemical sensitivity. We demonstrate that, through modulations of soft X-ray illumination in the near-field (Fresnel regime)[27,28], a "sample-as-optics" philosophy of PE-RSoXS leads to an efficient detection of electrochemical interfaces under *operando* conditions at the sub-nanometer scales beyond the soft X-ray diffraction limit. RSoXS integrates the structural sensitivity of small-angle X-ray scattering with the chemical specificity of near-edge X-ray absorption fine structure (NEXAFS). By tuning the incident photon energy near elemental absorption edges, RSoXS selectively enhances contrast based on chemical composition, molecular orientation, or electronic states of the materials. This element- and bond-specificity makes RSoXS particularly functional for probing heterogeneous systems, where subtle chemical variations are coupled to nanostructures[29,30]. Inspired by traditional crystallography, where periodicity in crystals enables coherent enhancement of scattering intensities[30,31], PE-RSoXS deploys engineered nanopatterns, with periods in the order of tens of nanometers, which act as optical elements manipulating the incoming wavefront in the near field (The near field here refers to the electromagnetic field that propagates through a nanostructure, calculated both within and around it, and influenced by the nanostructure itself.) and coherently enhancing scattering signals from every interface of the periodic structure, leading to sub-nanometer sensitivity. Unlike conventional approaches treating the sample as a passive object, PE-RSoXS incorporates the patterned nanostructures into the optical pathway, enabling coherent interference and resonant contrast enhancement[32] (Supplementary Table 1 and Supplementary Note 1).

PE-RSoXS provides an accessible approach for monitoring the evolution of catalysts at electrochemical solid-liquid interfaces under *operando* conditions. Ni line-grating nanopatterns (denoted as Ni LGNPs) were designed and fabricated as optical architectures for PE-RSoXS measurements during electrochemical water oxidation, a key reaction for green hydrogen production[33,34]. Structural dynamics such as lattice incorporation, dissolution/redeposition, and volume changes coexist with subtle chemical variations, like oxidation state shifts of active sites, all occurring within buried and often inaccessible environments[21,35,36]. PE-RSoXS captures these dynamics by integrating with rigorous finite-element simulations, leveraging the wave-like properties of soft X-rays, their spatial compatibility with interfacial phenomena, and the coherent amplification enabled by periodic nanopattern designs. This approach delivers substantial sensitivity and detection efficiency simultaneously for probing complex interfaces.

## Results and discussion
### Working principle of PE-RSoXS
PE-RSoXS approach is conceptually related to diffraction anomalous fine structure, a method noted by Cauchois et al.[37] and formalized by Stragier et al.[38], which combines Bragg diffraction with anomalous scattering near absorption edges to obtain element-specific structural information[39]. Well-ordered line-grating nanopatterns with uniform rectangular geometry were constructed to enhance scattering sensitivity. The angular spacing of Bragg spots reflects the pitch of the nanopatterns (structure factor), while the relative intensity distribution is governed by the line width of the nanopatterns (form factor). PE-RSoXS exploits its sensitivity to the form factor to reveal the buried electrochemical solid-liquid interfaces by enabling coherent amplification of resonant scattering signals, which effectively and uniquely decouples the interfacial signals from the response of the bulk material based on the diffraction order dependency (See details in Supplementary Notes 2 and 3). This makes PE-RSoXS sensitive to structural size changes with nanometer to sub-nanometer at the reaction fronts. Furthermore, by tuning the X-ray energy near the absorption edge of target elements, PE-RSoXS features site- and valence-state-specific capabilities comparable to XAS, providing an *operando* approach to track chemical evolution statistically during reaction processes[13,29,36,40,41].

In this study, we demonstrate the PE-RSoXS methodology by designing periodic Ni LGNPs for electrochemical water oxidation. As shown schematically in Fig. 1 and Supplementary Fig. 1, *operando* PE-RSoXS is performed in a customized electrochemical liquid flow cell with Ni LGNPs (see details in Methods). Through the SiN$_x$ window, the Ni LGNPs are probed by incident soft X-rays with the resonant photon energy near the Ni L-edge. The scattering signal from line gratings gives rise to an array of Bragg spots perpendicular to the orientation of the line-grating plane. The scattering results from the Ni LGNPs with 50% duty cycle structures show that the even-order diffraction intensity is significantly reduced or absent, while the odd-order signal remains prominent (Supplementary Note 4). Upon applying an anodic potential, the Ni line-grating develops a core-shell structure with an increase in width. PE-RSoXS allows a direct distinguishing of oxidized Ni shell from metallic Ni core. With the exposure time at the millisecond level and an incident X-ray beam dosage below 0.01 mJ cm$^{-2}$, this approach achieves the *operando* monitoring of structural and chemical dynamics of Ni LGNPs simultaneously at the solid-liquid interfaces when undergoing mild oxygen evolution reaction (OER) in the alkaline media. In our implementation, the interfacial region is defined as the shell layer in a Ni core-shell structure. This region evolves chemically and structurally during electrochemical OER, and its presence modifies the form factor, which is the basis for scattering contrast in PE-RSoXS. Unlike conventional scattering methods, where the signal scales linearly with the number of scatterers ($N$), PE-RSoXS utilizes the coherent interference between the scattered waves, resulting in the diffraction intensity scaling as $N$ squared[42]. In our experimental setup, the patterned window area was measured as approximately 80 μm × 80 μm, containing about 400 periodic units ($N \approx 400$). This coherent addition contributes to a five-order-of-magnitude increase in diffraction intensity compared to a single isolated scatterer (Supplementary Note 5), making PE-RSoXS well-suited for real-time monitoring of buried electrochemical interfacial dynamics with high efficiency. This signal amplification is proved by the electric effects in the near field (E-field) induced by incident soft X-rays (Fig. 1), which intensify the scattering signals and underscore the alignment between the scale of optical modulation and the physical dimensions of line-grating structures. By incorporating optical design principles directly into the electrocatalyst geometry, the sample itself becomes an active component of the measurement system, bridging structural modulation and spectroscopy to selectively probe

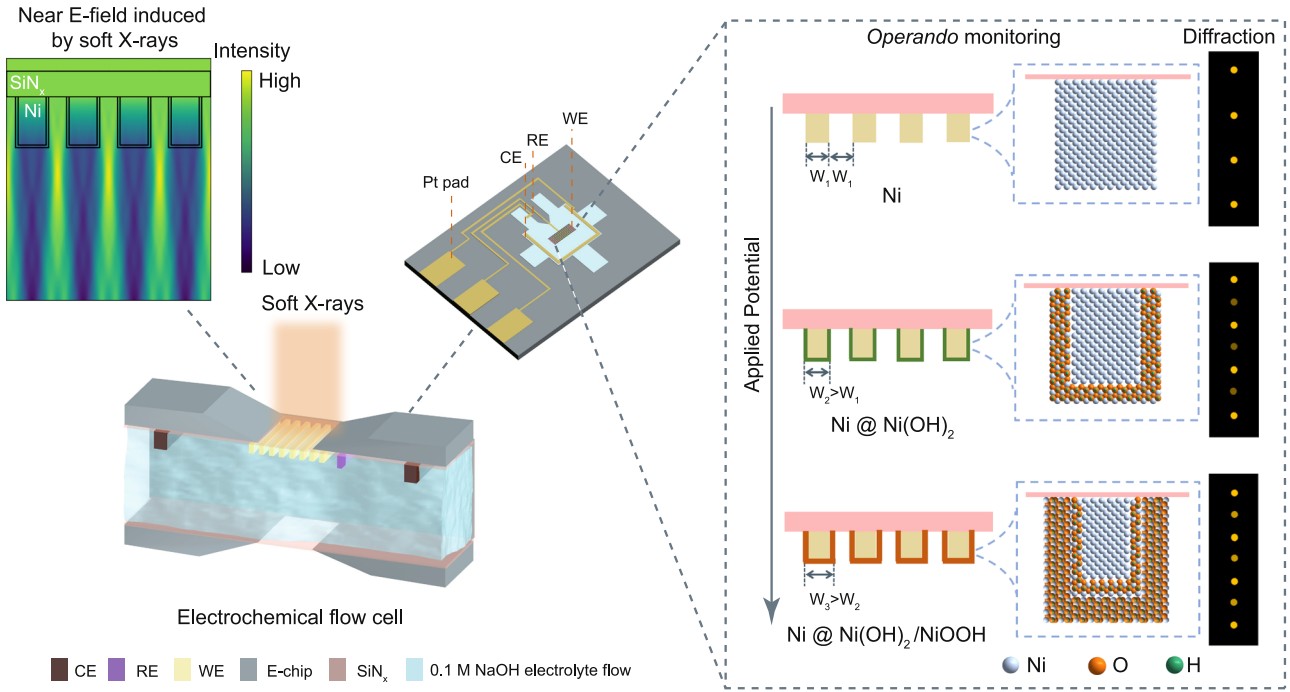

**Fig. 1 | Schematic illustration of the PE-RSoXS methodology.** The scheme shows the near E-field induced by soft X-rays for *operando* monitoring of solid-liquid interfaces during electrochemical water oxidation on Ni LGNPs. The abbreviations of CE, RE, WE, and E-chip correspond to counter electrode, reference electrode, working electrode, and electrochemical microchips.

electrochemical interfacial dynamics with high dimensional sensitivity and elemental resolution during operating conditions.

## *Operando* monitoring of electrochemical solid-liquid interfaces by PE-RSoXS

We fabricated well-ordered Ni LGNPs on the two SiN$_x$ windows of the sample E-chip using e-beam lithography, followed by Ni metal deposition and lift-off processes. These Ni line gratings exhibit 100 nm of width, 200 nm of pitch, and 50 nm of height (Fig. 2a). We employed Ni LGNPs as the WE integrated into a customized electrochemical liquid flow cell for OER in 0.1 M NaOH aqueous electrolyte. Cyclic voltammetry (CV) profiles first confirmed well-defined electrochemical behaviors of Ni LGNPs, consistent with measurements of a Ni metal foam performed in a standard H-cell (Supplementary Figs. 2 and 3). The electrochemical redox transitions of Ni/Ni(OH)$_2$ (0/+2) and Ni(OH)$_2$/NiOOH (+2/+3) are well understood and illustrated in the Bode diagram[43,44]. In our *operando* PE-RSoXS experiments, all the applied potentials were calibrated and converted to the reversible hydrogen electrode (RHE) based on our previous reports (Supplementary Fig. 3)[13,36]. All the potentials in this work were shown versus the RHE without iR correction unless otherwise noted. We applied potentiostatic scans at the open circuit potential (OCP), 1.2, and 1.6 V, respectively, which mainly aimed to monitor the chemical dynamics of electrocatalytic Ni sites before and within the OER potential range. The obtained three diffraction patterns were collected with 1 ms exposure time of soft X-rays at 852 eV, as shown in Fig. 2b. The Bragg spots corresponding to different diffraction orders were aligned, and the beam stop was positioned to block the direct beam and first orders. As shown in Fig. 2c, the scattering intensity of different orders varies at the applied potentials, extracted by summing the area around each peak over 60 pixels (px). A significant increase in scattering intensity at 1.6 V reflects the widening of the Ni line gratings, implying volumetric swelling of the Ni LGNPs under OER conditions[35,45]. The scattering intensities near the Ni L$_3$-edge, ranging from 849 eV to 861 eV were extracted and integrated as a function of photon energy at OCP, 1.2

and 1.6 V, respectively, producing energy-dependent scattering profiles. While these spectra resemble XAS in that they exhibit Ni L-edge features, they are not directly equivalent. The measured diffraction intensities result from spatially modulated differences in the complex refractive index ($n = 1 - \delta + i\beta$) and reflect both absorption ($\beta$) and dispersion ($\delta$) contributions. These scattering-derived spectra were resolved with an exposure time of 1 ms for each energy point, and the irradiation intensity of the incident X-ray beam was kept below 0.01 mJ cm$^{-2}$. The spectra, derived from the diffraction intensities of the 3rd (Fig. 2d), 4th (Fig. 2e), 5th (Fig. 2f), and 6th orders (Supplementary Figs. 4–6 and Supplementary Notes 6 and 7), illustrate the distinct spectroscopic features of the Ni LGNPs[46,47].

To connect the scattering spectroscopic features with the chemical and structural dynamics at the electrochemical solid-liquid interfaces, we performed the simulations to model the Ni LGNPs under realistic reaction conditions using a finite-element method (FEM) based Maxwell solver[48,49]. The rigorous FEM modeling comprehensively accounts for multiple scattering and near-field interactions, which are essential for analyzing sub-wavelength structures and materials. The FEM-based simulation details are provided in Methods. The structural and chemical dynamics at different potentials were clarified by performing the energy-dependent optical constant simulations of the core and shell of the Ni LGNPs and thus obtaining the simulated scattering-derived spectra to align with our experimental PE-RSoXS results. The corresponding inverse problem was addressed with a Bayesian optimizer (BO), a machine learning tool that calculates the best next sampling point by integrating the expected improvements[50]. The finite elements were indicated by small triangles in the models (Fig. 2g). In our simulations, we modeled the electrolyte environment as pure water, instead of the 0.1 M NaOH aqueous solution in the experiments, to simplify the computational framework. The presence of NaOH primarily influences the ionic strength and pH of the solution but has an insignificant effect on the scattering contrast at the Ni L-edge. This is because Na$^+$ and OH$^-$ ions have no absorption features in the energy range of Ni L-edges, nor do they significantly alter

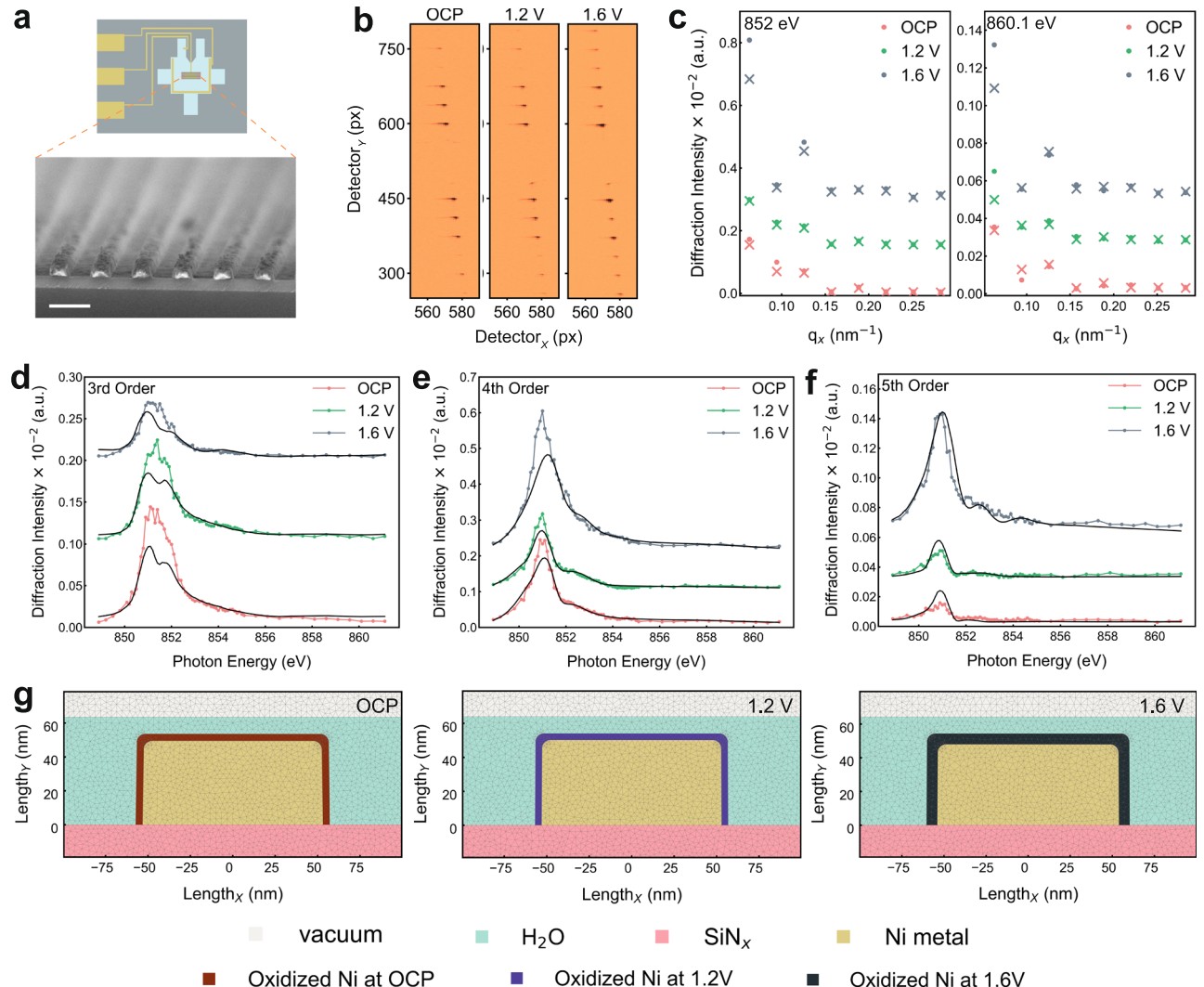

**Fig. 2 | *Operando* probing of solid-liquid interfaces on Ni LGNPs using PE-RSoXS. a** Schematic illustrations of the sample chip supporting the three-electrode circuit in the liquid flow cell, along with the corresponding scanning electron microscopy (SEM) image of the Ni LGNPs on the SiN$_x$ windows on the sample chip. Scale bar, 200 nm. **b** Diffraction patterns from charge-coupled device (CCD) images measured at open circuit potential (OCP), 1.2 and 1.6 V at 852 eV of the photon energy, and **c** the corresponding diffraction intensity profiles (solid round points) measured at 852 and 860.1 eV, respectively, with the fitting results (crosses). Scattering-derived spectra from the **d** 3rd, **e** 4th, and **f** 5th orders measured at OCP, 1.2 and 1.6 V, respectively, near the Ni L-edge with simulation results based on the finite-element-method (FEM) models. **g** Cross-sectional models of the gratings for FEM simulations at OCP, 1.2 and 1.6 V, with the color maps of the components in the models. All the electrochemical measurements were conducted using 0.1 M NaOH aqueous solution flow with a flow rate of 250 μL h$^{-1}$ at the room temperature without iR correction.

the dielectric function of the electrolyte at these photon energies[32,46,51,52]. Three core-shell models represent the Ni LGNPs at the three applied potentials. The models retain the same metallic Ni core, while the chemical compositions and widths of the interfacial shells vary. While our FEM-based simulations provide an agreement with experimental data across multiple diffraction orders, the models assume idealized core-shell structures with sharp, uniform interfaces. Interfacial roughness, line-edge disorder, and thickness inhomogeneity, especially common in solid-state electrochemical systems, may reduce the intensity of higher-order diffraction peaks and influence parameter extraction. These effects are partially accounted for using a Debye–Waller factor (see Methods), but residual mismatch or misfit could still arise from structural imperfections not explicitly modeled (Supplementary Fig. 7 and Supplementary Note 8).

It is clear that the scattering-derived spectra of different orders at OCP and 1.2 V are comparable, which suggests a negligible change of chemical composition at the reaction fronts without a major alteration in its overall structure by increasing the potential and keeping it before

**Table 1 | The featured parameters of Ni LGNP models in FEM-based simulations at different potentials with confidence intervals from Eq. 7 and modeling residual $\chi^2$ from Eq. 4**

| Potential (V vs. RHE) | Core width (nm) | Shell thickness (nm) | Total width (nm) | $\chi^2$ |
|---|---|---|---|---|
| OCP | 105.6 ± 0.27 | 4.1 ± 0.11 | 113.8 ± 0.49 | 5.6 |
| 1.2 | 105.6 ± 0.21 | 3.9 ± 0.07 | 113.4 ± 0.35 | 6.1 |
| 1.6 | 107.7 ± 0.04 | 6.4 ± 0.11 | 120.5 ± 0.26 | 7.5 |

The total width equals the sum of the core width and twice the shell thickness.

OER. By contrast, a clear transformation of the structure and chemical composition was captured at 1.6 V under the OER condition. Details of the model parameters are provided in the Methods section, and the features of Ni LGNPs at different potentials are summarized in Table 1. To clarify the structural and chemical changes induced by different potentials, the energy-dependent optical constants ($n(E) = 1 - \delta(E) + i\beta(E)$) simulations of the core and shell of the Ni LGNP models are

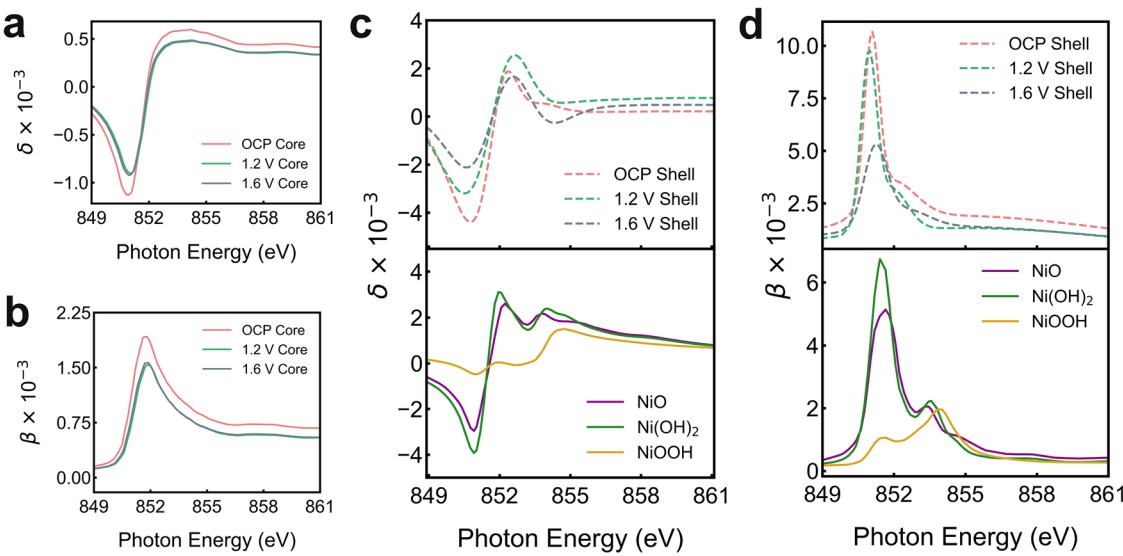

**Fig. 3 | Simulation of the optical constants for the core and shell of Ni LGNP models at different potentials. a**, **c** Dispersion component, $\delta(E)$ and (**b**, **d**) absorption component, $\beta(E)$ of the complex refractive index ($n(E) = 1 - \delta(E) + i\beta(E)$) of the core and shell of Ni LGNPs at OCP, 1.2 V and 1.6 V. Reference spectra for NiO, Ni(OH)$_2$, and NiOOH are included for comparison as a function of photon energy near the Ni L-edge absorption.

presented in Fig. 3. The real ($\delta(E)$) and imaginary ($\beta(E)$) parts of the optical constants are illustrated for the core of the Ni line gratings in Fig. 3a, b, respectively, remaining largely unchanged with applied potentials and preserving the spectral fingerprint of metallic Ni in the bulk. This indicates the electrochemical stability of the line-grating core. The optical constants of the line-grating core, derived from NEXAFS and the Kramers–Kronig (KK) relation[53,54], were fitted using an energy offset and density scaling. Meanwhile, $\delta(E)$ and $\beta(E)$ for the shell are exhibited in Fig. 3c, d, respectively, alongside comparisons to reference spectra for NiO, Ni(OH)$_2$, and NiOOH. For $\delta(E)$, indicating the structural changes, the results are apparently consistent, referring to the simulations of the references, suggesting the subtle width changes of the Ni LGNPs (around 2 nm) during electrolysis and thus highlighting the structure sensitivity of PE-RSoXS. For $\beta(E)$, which corresponds to the absorptive component of the optical response, the simulation captures key features of Ni L-edge absorption in the shell region. These features are not direct XAS measurements but rather arise from the fitted optical constants that reproduce the energy-dependent scattering intensity across all diffraction orders. The simulated results for the shell exhibit higher absorption at around 851.5 eV but lack of the shoulder near 854 eV at OCP and 1.2 V, suggesting Ni$^{2+}$ was the dominant state in the shell at both potentials. However, at 1.6 V, the optical constants resemble a mixture of Ni(OH)$_2$ and NiOOH, as the absorption profile aligns closely with a mixture of Ni(OH)$_2$ and NiOOH, yet it is overall lower than those at OCP and 1.2 V. When the OER was actively driven, the main peak intensity decreased at around 851.5 eV while a distinct shoulder at around 854 eV emerges, apparently matching the spectral features of Ni$^{3+}$ in γ-NiOOH. This transition features the in-situ formation of a higher-valence Ni$^{3+}$ phase, which is widely regarded as the catalytically active species for OER[34,55,56]. The evolution of the shell (interfacial) region is modeled as a change in form factor due to the oxidation-state-dependent optical constants. The observed spectral changes are not exclusive to interfaces but rather reflect the local chemical environment within the shell region defined in the form factor.

Based on the simulation results, we captured the potential-dependent evolution of a model Ni core-shell geometry, where the core remains metallic Ni, and the shell represents the dynamic interfacial layer undergoing oxidation transitions. As shown in Table 1, upon increasing the potential from OCP to 1.2 V, the Ni LGNPs exhibited a sub-nanometer lateral contraction (-0.4 nm) in total width. The spatial resolution of PE-RSoXS is fundamentally limited by the X-ray wavelength (-1.5 nm), resulting in a diffraction-limited resolution of approximately 0.75 nm. Our ability to detect smaller structural changes (e.g., 0.2 nm in shell thickness) arises not from direct imaging beyond this limit, but from the sensitivity of diffraction intensities to such variations within the periodic structure. This dimensional change could be attributed to early-stage surface oxidation and phase transitions, leading to denser surface phases and potential-induced lattice strain[57,58]. Meanwhile, the extracted geometric parameters reveal that the shell thickness increases from 4.1 nm at OCP to 6.4 nm at 1.6 V. This shell thickness modulation, accompanied by a progressive increase in total width and modeling residual ($\chi^2$), reflects an oxidation-driven interfacial reorganization under anodic bias[35,45]. Importantly, the fidelity of the simulations in reproducing experimental scattering-derived spectra across multiple diffraction orders underscores the high sensitivity of our PE-RSoXS technique to local chemical environments and subtle changes of structures during electrocatalysis. The similarity in scattering-derived spectral features between OCP and 1.2 V suggests that the initial stages of electrochemical polarization do not significantly alter the interfacial structure statistically, despite partial oxidation of surface species. In contrast, the clear evolution of spectral features at 1.6 V indicates a nontrivial transformation of the interfacial shell layer, coinciding with the onset of OER activity on Ni sites. The increase in shell width and the emergence of Ni$^{3+}$ spectral signatures together point to an electrochemically driven reconfiguration of the solid-liquid interfaces that enhances catalytic functionality (Supplementary Note 9). Besides, our simulated total width changes of the Ni LGNPs align with atomic force microscopy (AFM) results from the samples after being operated at the corresponding conditions (Supplementary Fig. 8), further clarifying the fidelity of our FEM-based simulations. However, in these AFM images, clear fluctuations are observed along the line edges. Ex situ microscopy techniques cannot directly measure the oxide layer thickness with sub-nanometer sensitivity. In contrast, by exploiting resonant scattering and diffraction-order sensitivity, PE-RSoXS can distinguish shell and core contributions without relying on post-mortem assumptions, providing statistically representative results across hundreds of nanopatterns under realistic electrochemical conditions, which highlights the significance to develop PE-RSoXS.

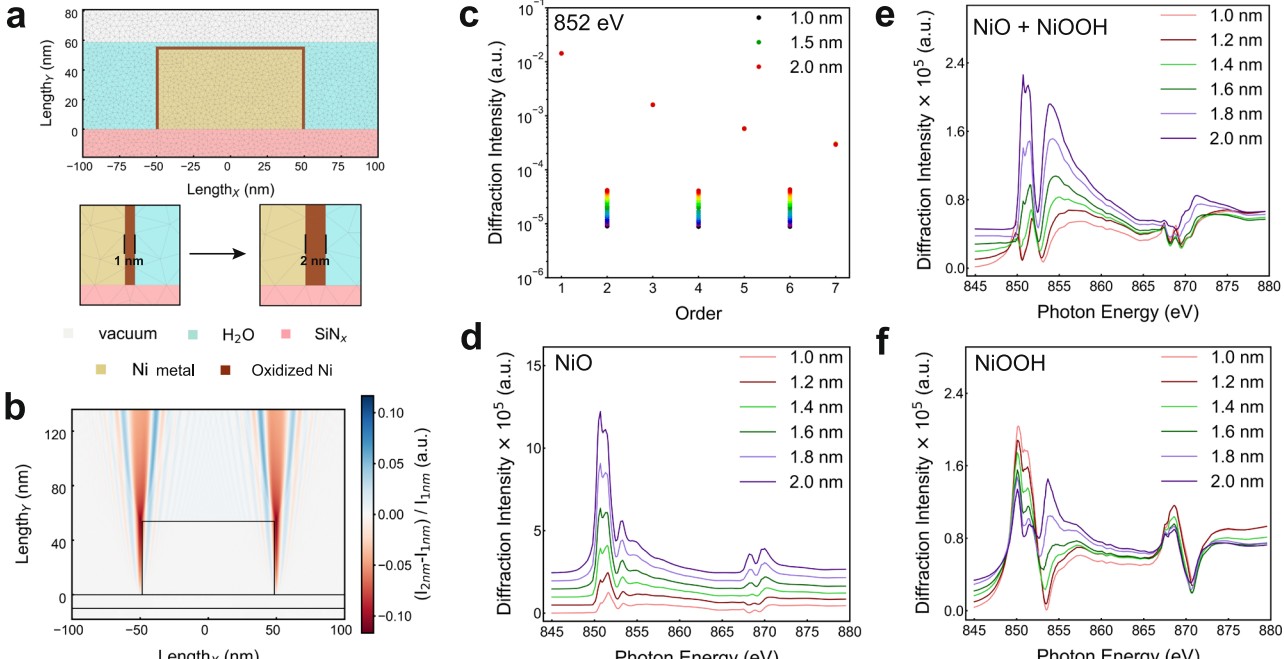

**Fig. 4 | Theoretical FEM-based simulations of PE-RSoXS. a** A FEM model of the Ni LGNP showing a rectangular core-shell nanostructure with a Ni metal core and an oxidized Ni shell. The width of the shell increases from 1 to 2 nm in the simulations. The color maps of components in the models are provided. **b** Relative intensity simulation of the near E-field induced by soft X-rays at photon energies of 880 eV. **c** X-ray diffraction intensities of varied diffraction orders based on the models of metallic Ni in the core and NiO in the shell as the width of the NiO shell increases by 1 nm at 852.0 eV photon energy. Spectral variations of the 2nd order as the shell width increases by 1 nm for the models with the shell of (**d**) bare NiO, (**e**) a mixture of NiO and NiOOH, and (**f**) bare NiOOH, respectively.

To further assess the dimensional sensitivity of PE-RSoXS, we monitored the dynamic redox kinetics of Ni LGNPs during potential cycling (Supplementary Fig. 9). By altering the potential between 0.8 and 1.4 V, we observed dynamic oscillations with three distinct scattering peaks corresponding to oxidation and subsequent reduction in the reversible potential scans. Both $\alpha$-Ni(OH)$_2$ and $\gamma$-NiOOH crystals, which usually form under OER conditions, exhibit layered structures with interlayer water, resulting in d-spacings of approximately 8 and 7 Å, respectively. In contrast, the unhydrated $\beta$-Ni(OH)$_2$ and $\beta$-NiOOH phases have significantly smaller interplane d-spacings of 4.6 and 4.8 Å, respectively[59]. The phase dynamics between $\alpha$-Ni(OH)$_2$ and $\beta$-Ni(OH)$_2$, or between $\beta$-NiOOH and $\gamma$-NiOOH, as well as preliminary redox between $\beta$-Ni(OH)$_2$ and $\gamma$-NiOOH, would theoretically lead to the changes of the line-grating width due to lattice swelling and shrinking. Therefore, the fluctuations in the 2nd-order intensity, compared to the stable 3rd-order intensity, could likely be attributed to the phase changes and redox processes of Ni LGNPs[60].

These results unveil the distinctive capability of PE-RSoXS to *operando* resolve subtle structural and chemical transformations at buried solid-liquid interfaces. The method quantitatively captures sub-nanometer dimensional changes in the Ni LGNPs during OER while simultaneously mapping oxidation-state gradients between the metallic Ni core and the evolving Ni(OH)$_2$/NiOOH shell through the FEM simulations on the diffraction-order-dependent scattering spectra. This dual sensitivity to geometry and chemistry, achieved under low X-ray doses with statistical averaging across hundreds of identical nanopattern units, enables a direct correlation between redox transitions and sub-nanometer morphological reconstruction. Such interface-specific insights, particularly the separation of bulk and interfacial signals based on the core-shell structures, are essential for understanding electrocatalytic behaviors, which is still difficult to obtain with conventional soft XAS or electron microscopy under realistic electrochemical conditions.

## Theoretical FEM-based simulations of PE-RSoXS

Building on the alignment between experimental results and simulations, we further conducted theoretical FEM-based modeling of PE-RSoXS to elucidate its potential as a sensitive probe for nanostructural and compositional variations. Uniform and rectangular core-shell line-grating nanopatterns with 100 nm width and 50 nm height (Fig. 4a), where Ni metal was in the core and oxidized Ni species were in the shell, were built up on a SiN$_x$ substrate to mimic the scattering behaviors. Under electrochemical conditions, our theoretical FEM-based simulations of PE-RSoXS remain robust and representative of the Ni LGNPs. The optical constants of all compositions were employed for the simulations at the corresponding incident energy range (Supplementary Fig. 10), of which $\beta(E)$ as a function of photon energy, the absorption component, was calculated from NEXAFS measurements and simulations; while $\delta(E)$ as a function of photon energy, the dispersion component, was derived from the KK relations. Diffraction intensity is primarily determined by the contrast between the line gratings and the environmental liquid. We simulated the near E-field intensity induced by soft X-rays by increasing the shell width from 1 to 2 nm and mapped out the relative E-field intensity (($I_{2\,nm}-I_{1\,nm}$)/$I_{1\,nm}$), which exhibits the modulation of soft X-ray illumination by the engineered nanopatterns as optical elements in the Fresnel regime. As shown in Fig. 4b, a clear enhancement of the intensity contrast by the shell width variation can be observed at the interfaces. Importantly, the optical modulation scale for E-field enhancement closely aligns with the length scale of interfacial shell structures, significantly improving the measurement sensitivity to the electrochemical interfacial dynamics via PE-RSoXS.

At the resonant photon energy (i.e., 852.0 eV, the Ni L-edge energy), the diffraction intensities were obtained in the direction of the incident X-ray perpendicular to the nanopattern plane. As shown in Fig. 4c and Supplementary Fig. 11, the odd-order intensities remain unchanged. Strikingly, those of the even orders exhibit apparent enhancement induced by statistically identical interfaces of oxidized Ni with only the width variations from 1 to 2 nm by 0.2 nm of a step.

Meanwhile, site-specific chemical composition was identified by order-dependent spectroscopy derived from the scattering intensities near the Ni L-edge energy. By changing the width of oxidized Ni interfaces, the odd-order spectra show the same feature of Ni metal core (Supplementary Fig. 12), which elucidates the bulk state of the Ni LGNPs. In contrast, the even-order spectra depict an increasing signature of NiO with the increase of the shell width (Fig. 4d and Supplementary Fig. 12), indicating that the even-order spectra reveal the interfacial Ni oxidation states. We further extend the chemical composition of the oxidized Ni shell from bare NiO towards the mixture of NiO and NiOOH (Fig. 4e), as well as bare NiOOH (Fig. 4f). The involvement of $Ni^{3+}$ species enhances the spectral features at around 854 eV (Supplementary Fig. 12), compared to that of the $Ni^{2+}$ state, highlighting the considerable chemical sensitivity of PE-RSoXS[61]. To address the concept of spatial resolution more rigorously, we conducted additional simulations (Supplementary Fig. 13). The oxidized Ni shell region was displaced with distances from the Ni metal core without physical contact. These simulations reveal that the diffraction signal is not only sensitive to shell thickness and composition, but also to the position of the shell within the grating period, confirming that PE-RSoXS can resolve spatial placement within the structural unit cell. By simulating unphysical scenarios in which the shell is "floating" at varying positions within the groove, we demonstrate that the diffraction intensities change depending on shell placement, indicating the system's sensitivity to spatial localization of chemical features.

As a result, our simulations theoretically demonstrate the characteristic features of PE-RSoXS in unveiling interfaces: (i) enhanced scattering signal from buried interfaces by statistically identical nanopatterns (form factors) and decoupled the interfacial signals from the bulk material responses; (ii) high scattering sensitivity to sub-nanometer structural variations; (iii) site-specific sensitivity of elemental oxidation state by order dependency; (iv) achievable spatial resolution within the unit cell when the geometry is well-defined.

In summary, we report the development of *operando* PE-RSoXS, which is a powerful and non-destructive characterization methodology for probing dynamic solid-liquid interfaces with sub-nanometer structural sensitivity and millisecond temporal resolution. Deployed via a "sample-as-optics" design, PE-RSoXS incorporates nanoengineered periodic structures directly into electrochemical samples, transforming them into diffractive optical elements that coherently amplify faint resonant scattering signals from buried interfaces. This architecture leverages the length-scale match between soft X-ray wavelengths and chemical structures, offering elemental specificity and spatial precision. Using a simulation framework based on electromagnetic wave propagation through structured media, we quantitatively resolve oxidation-state gradients and subtle structural changes on Ni LGNPs under *operando* OER conditions. Diffraction-order-dependent contrast enables clear separation of interfacial dynamics from bulk signals, allowing real-time probing of chemical and structural evolution through form factor sensitivity to chemical distributions within the unit pattern. The approach is not fundamentally limited to perfectly periodic systems. One potential strategy is to use a patterned template and apply a porous catalyst on top, leveraging the underlying structure to enhance diffraction signals. This capability facilitates mechanistic insights into catalyst structure-function relationships, directly informing the design of next-generation electrochemical materials.

PE-RSoXS introduces an interface-specific metrology by combining resonant soft X-ray contrast with Bragg diffraction enhancement from tailored nanopatterns. The PE-RSoXS methodology is broadly applicable beyond the Ni-based system studied in this work, providing *operando* access to coupled structural and chemical dynamics in a wide range of catalytic materials that undergo significant restructuring during operation. By tuning to the relevant absorption edges of the focused elements, PE-RSoXS can selectively enhance scattering signals from specific chemical states, enabling real-time correlation between

sub-nanometer structural changes and active-phase evolution. Its compatibility with identical architecture, including powder- and pattern-based systems, low-dose operation, and high sensitivity to the buried electrochemical interfaces make it a versatile tool for probing diverse catalysts under realistic electrochemical conditions, with strong potential to address key questions in catalysis, energy conversion, and materials science. Moreover, by coherently amplifying signals from statistically identical units, PE-RSoXS is compatible with broadband (pink beam) illumination and enables single-shot, full-spectrum acquisition, dramatically improving throughput and reducing acquisition time. This opens exciting opportunities for coupling with ultrafast detectors, diffraction-limited synchrotron sources, and X-ray free-electron lasers, paving the way for real-time tracking of ultrafast interfacial transformations under realistic operating conditions.

## Methods

### Materials
The commercial Poly(methyl methacrylate) (PMMA) photoresist was purchased from Polymer Source, Inc. Acetone (SKU No. 270725, 99.9%), Ethanol (SKU No. 459844, 99.5%), Ni metal (SKU No. 767484, 99.95%), NaOH (SKU No. 306576, 99.99%) and sulfuric acid ($H_2SO_4$, SKU No. 258105, 95-98%) were purchased from Sigma-Aldrich to be used in this work without further purification. The Ni foam (SKU No. bcnf-16m, 99.9 wt.%, 1.6 mm thickness), graphite rod electrode (SKU: EFC-CE-G), Ag/AgCl reference electrode (SKU: EFC-RE-AAC), and standard electrochemical H-cell (SKU: EC-HDC-GB-GFRS-50, 50 mL volume) were purchased from MTI corporation. Nafion N117 proton exchange membrane (177.8 μm thickness) was obtained from Ion Power, Inc. Customized electrochemical microchips (E-chips) were obtained from Protochips, Inc. as X-ray transmissive substrates with two $SiN_x$ windows (240 μm in length and 80 μm in width). Milli-Q water (resistivity 18.2 MΩ cm at 25 °C) was used for the preparation of the 0.1 M NaOH aqueous solution.

### Nanofabrication of Ni line-grating nanopattern electrodes
Ni line-grating nanopatterns were fabricated on the $SiN_x$ windows of the E-chips using e-beam lithography, followed by Ni metal deposition and lift-off processes at the Molecular Foundry's Nanofabrication facility of Lawrence Berkeley National Laboratory. The E-chips were first rinsed with acetone, ethanol, and isopropyl alcohol sequentially, then cleaned under UV-ozone for 6 min. Then E-chips were dehydrated in an oven at 150 °C for 5 min and then cooled down to room temperature for 10 min. The E-chips were then spin-coated with around 180 nm-thick PMMA photoresist at 2000 rpm, and oven-baked for another 5 min. Next, the e-beam lithography was used to expose the line gratings with a width and pitch of 100 and 200 nm, respectively, to cover the $SiN_x$ windows and connect the Pt circuit of the working electrode. Kapton tape was applied to the areas outside the vicinity of the patterned working electrode region before e-beam evaporation in order to minimize the likelihood of unintentional shorts between the counter or reference electrodes and the deposited Ni metal. A 50-nm height of Ni metal was deposited by e-beam evaporation and then lifted off by stripping the PMMA with acetone. The cross-section image of Ni LGNPs was captured with SEM imaging (ZEISS) at 5 kV.

### *Operando* PE-RSoXS measurements
A customized electrochemical liquid flow cell constructed by Protochips, Inc. was developed at the beamline 11.0.1.2 of the Advanced Light Source in Lawrence Berkeley National Laboratory[62]. RSoXS patterns were collected with a back-illuminated Princeton PI-MTE charge-coupled device (CCD) cooled to −45 °C. Each RSoXS pattern at one energy point was acquired in a single 1 ms exposure time of soft X-rays using 1/1000 of the incident photon flux (X-ray dosage per shot: below 0.01 mJ cm$^{-2}$). This ultralow-dose configuration enables highly sensitive detection of scattering signal changes of the line gratings while preserving the native state of the electrochemical interfaces. An E-chip from

Protochips, Inc. with Ni LGNPs was paired with a small supporting chip with a 500-nm spacer for the encapsulation of 0.1 M NaOH aqueous electrolyte. Both the counter and reference Pt electrodes were integrated on the E-chip. 0.1 M NaOH aqueous solution was prepared freshly before every electrochemical measurement using Mili-Q water with stirring. The pH value of the electrolyte was confirmed as $12.96 \pm 0.24$ by a pH meter purchased from METTLER TOLEDO (SevenDirect SD20). The distance between the reference electrode (RE) and the working electrode (WE) is minimized to reduce the iR-drop within a 500-nm-thick electrolyte, while the counter electrode (CE) has an extensive surface area to facilitate rapid polarization. Electrochemical resistance measurements or iR correction were not intentionally performed in this study, as our focus lies on exploring the electrochemical reactions during water oxidation rather than optimizing the system's electrocatalytic performance. However, to confirm the integrity of our electrochemical cell setup, the resistance was measured to be around 20 kΩ by single-frequency electrochemical impedance spectroscopy at 10,000 Hz. Then, the same electrolyte was flowed continuously, facilitated by a Harvard Apparatus PHD ULTRA Programmable Syringe Pump with a rate of 250 μL h⁻¹, minimizing the bubble retention effect. Electrochemical measurements were performed with a Gamry potentiostat in a 3-electrode floating mode configuration. The data was collected and processed by the software, Gamry framework. Prior to constant-potential PE-RSoXS measurements at OCP, 1.2 V, and 1.6 V vs. RHE, the Ni LGNP working electrodes underwent a standardized conditioning sequence to stabilize the electrochemical interfaces. Firstly, as-fabricated Ni LGNPs were characterized by RSoXS in the absence of electrolyte to establish baseline RSoXS patterns. Next, several CV scans were performed over a potential range from OCP to 1.2 V vs. RHE, covering the Ni oxidation/reduction region, to remove surface impurities, activate the nanopatterns, and establish a stable electrochemical response. This typical method ensures reproducible electrocatalytic activity and RSoXS patterns by removing adventitious surface species and stabilizing the core-shell structure prior to operando measurements. Following conditioning, scattering profiles at the Ni L-edge were collected operando under constant-potential control at OCP, 1.2 V, and 1.6 V vs. RHE. The topographic imaging of the nanopatterns after operating at the three potentials was captured using AFM, which was conducted using the Bruker Dimension Icon3 in ScanAsyst mode at the Molecular Foundry of Lawrence Berkeley National Laboratory.

## Theoretical FEM-based simulations

For a quantified simulation of the PE-RSoXS measurements, we constructed and simulated Ni line-grating nanopatterns using a FEM-based Maxwell solver from JCMwave. The first step for the calculation was to define a model. All the shapes were polygons (especially trapezoids), and the coordinates were generated from the following parameters: side-wall angle, height, and width by Python scripts. The model contained the computational domain and defined the maximum size of the simulated cross-section based on the total height of the line gratings, the height of the SiN$_x$ window, and an offset above and below the structure. The width of the line gratings was defined by the pitch. The boundary conditions of the computational domain were classified for the solver: the upper and lower boundaries (Y-direction) were transparent, while the right and left (X-direction) boundaries were periodic. The SiN$_x$ window was set in a thickness of 50 nm, and a line grating with varied height, width, and side-wall angle was placed. One shell layer was added to the grating core. Additionally, the grooves of the grating were filled with water. The generation of the mesh comprised of the finite elements was completed based on the above settings. Smaller features were adapted by the chosen mesh compared to the side length constraints of 0.7 nm. All the polygons were connected to the selected materials with pre-defined optical constants ($n(E) = 1 - \delta(E) + i\beta(E)$) from the Henke database[63] except for the core and shell. For the core, XAS spectra were obtained with total electron yield. The absorption coefficient $\beta(E)$ in the optical constant was obtained by scaling the XAS with the atomic spectrum from the linear combination of each individual atom from Henke data, according to the chemical formula. The dispersion coefficient $\delta(E)$ was derived from $\beta(E)$ via KK relations. For the shell, the preliminary calculations were conducted by optical constants derived from reference spectra (Fig. 4), retrieved in a manner similar to those of the core. In subsequent modeling, these optical constants were approximated using Gaussian profiles in Fig. 2 and Supplementary Fig. 4. For the FEM-based simulations, the X-ray source was defined: an S-polarized X-ray light source with a varied wavelength. We established an incident plane wave coming below the SiN$_x$ window perpendicular to the plane of the line gratings, along with the orientation of the line width corresponding to the experimental configurations. The FEM-based Maxwell solver calculates the electric field intensity in the near field. To gain the scattering intensities of the diffraction orders, the far field is calculated with a Fourier transformation of the E-Field intensity distribution (near field). Because of the defined periodicity, the discrete scattering diffraction intensity with its **q**-space coordinates was retrieved as the simulation outputs. All the FEM-based calculations were conducted on the supercomputer in the National Energy Research Scientific Computing Center of Lawrence Berkeley National Laboratory by using the commercially available software package JCMsuite (Version 6.2; JCMwave GmbH, Germany).

## Experimental PE-RSoXS spectra fitting via simulation

To model the experimental PE-RSoXS results using FEM-based simulations, the experimental data were first reduced from 2D detector images into spectra for each diffraction order. The first step involves performing a background correction. To achieve this, the portion of the image (200 pixels wide), which does not contain any scattering intensities, was selected. We calculated line-wise in the X-direction, which was subsequently subtracted from the rest of the image to remove background interference. Following the background correction, the peaks were identified and labeled according to their diffraction order. Initially, the two peaks with the highest intensity were located using a peak-finding algorithm. A line was drawn through these peaks, and the intensities along this line were summed in the X-direction using a width of 60 pixels. The line cut was then taken, and the peak positions were identified. These positions remained stable during a single measurement, making the process robust and consistent. Once the peaks were identified, they were integrated over a width of 23 pixels. This process was repeated for each CCD image taken during the measurement. Additionally, an $I_0$ calibration was applied to the data to account for intensity variations in the beamline as a function of photon energy. The next phase involved parameterizing the model that fits the experimental data. The basic shape of the model was described in the previous section. The grating was represented as a rectangle, which was parameterized by its height, width, and side-wall angle. For the oxide layer, an additional thickness was incorporated in all directions onto the rectangle of the core. The optical constants of the core and shell were fitted, while the core was based on optical constants retrieved from NEXFAS and KK with a density and energy offset. The energy shift (−0.3 eV) and density scaling factor (0.55) for 1.2 V and 1.6 V were similar, while OCP showed a higher density (0.68) and a smaller energy shift (−0.2 eV). These differences were relatively minor, especially compared to those observed in the shell or oxide layer. For the shell, a more free-floating approach was chosen to account for compositions that are not able to be retrieved from other measurements (i.e., NEXFAS). Preliminary tests revealed that 8 Gaussians were required to describe the refractive index of Ni oxide, as retrieved from NEXAFS in the range of 840–865 eV.

$$y(E) = \sum_{i=1}^{4} a_i e^{-\frac{(E-E_i)^2}{2\sigma_i^2}} \tag{1}$$

$a_i$: amplitude of the Gaussian, $E$: energy, $E_i$: position of the Gaussian, and $\sigma_i$: width of the Gaussian.

The energy offset accounts for small calibration differences between the incident photon energy at the RSoXS beamline and the reference NEXAFS data used to compute the optical constants. This typically ranged between −0.2 and −0.3 eV and was used to align characteristic absorption features of the core. The shell thickness was treated as a floating parameter and optimized independently for each potential condition using Bayesian optimization. It varied along with the shell optical constants, Debye–Waller roughness, and core energy offset.

To account for the minimal measurable scattering intensity (dark current), a constant offset was added to the calculated intensity. Roughness was incorporated using a Debye–Waller factor correction[64]:

$$I_{fit}(K_x) = I_{calc}(K_x) * e^{-(K_x * \sigma_{rough})^2} \quad (2)$$

$I_{calc}(K_x)$: Calculated intensity in the diffraction order ($K_x$) with the FEM solver, $K_x$: Position of the diffraction spots in K-space, and $\sigma_{rough}$: Debye–Waller correction for roughness.

Roughness was incorporated using a Debye–Waller factor correction to model the attenuation of scattering intensity at higher diffraction orders due to structural disorder. However, this simplified approach does not fully account for the realistic effects such as lateral inhomogeneity, line-edge roughness, or non-uniform shell thickness. These imperfections could lead to suppression of higher-order peaks or fitting ambiguity and represent a source of model non-uniqueness.

The following error model was used for the Bayesian optimization method (BO) calculations[50]:

$$\sigma^2(E, \mathbf{q}) = \left(a I_{exp}(E, \mathbf{q})\right)^2 + b^2 \quad (3)$$

With $a = 0.05$ and $b = 10^{-5}$, which was a feasible approximation of the experimental error.

By optimizing the $\chi^2$, the parameters of the error are modeled based on the following equation:

$$\chi^2 = \sum_{E, K_x} \left( \frac{\left(I_{exp}(E, K_x) - scale\, I_{fit}(E, K_x)\right)^2}{2\sigma^2(E, K_x)} \right) \quad (4)$$

A total scale of 36 model parameters were optimized using the BO. Domain boundaries were determined based on expected model parameters derived from SEM images and NEXAFS spectra for the optical constants of NiO, Ni(OH)$_2$ and NiOOH. Bayesian optimization enabled efficient parameter sampling and paired well with the computationally intensive FEM solver.

To estimate the confidence intervals, we employed the method reported previously[65]. This involved inverting the Hessian matrix $H_{ik}$[66] of the error function:

$$H_{ik} = \frac{\partial \chi^2(\mathbf{p})}{\partial p_i} \frac{\partial \chi^2(\mathbf{p})}{\partial p_k} \quad (5)$$

where $\mathbf{p}$ represents the model parameters used for fitting.

At the minimum of $\chi^2$, we assumed the parameters followed a Gaussian distribution. The standard deviation (STD) at the global minimum parameter set was defined as:

$$STD = \sqrt{\frac{\chi^2}{DoF}} \quad (6)$$

where ($DoF = N−M$) represented the degrees of freedom with $N$ as the number of measurement points and $M$ as the number of free model parameters.

The confidence interval for each model parameter could be calculated as:

$$\sigma_{\mathbf{p}} = STD \sqrt{diag((\mathbf{H})^{-1})} \quad (7)$$

Utilizing the Gaussian process model for ($\chi^2(\mathbf{p})$) at its minimum allowed us to approximate the error covariance matrix, $(\mathbf{H})^{-1}$, and determine the confidence intervals of the parameters.

To improve smoothness and convergence, the fitting process was seeded using the optimized solution at the previous energy step as the initial guess for the next energy point. The optical constants $\beta(E)$ were parameterized using a sum of Gaussian functions, and $\delta(E)$ was calculated via KK relations, ensuring that both components followed physically meaningful trends. This regularization mitigated local minimum and produced smooth, interpretable spectral features.

### Standard electrochemical measurements in an H-cell

Bench-top electrochemical measurements were performed in a three-electrode H-cell using a Biologic potentiostat without iR correction. Before the electrochemical measurements, the bulk Ni foam was washed by 0.5 M H$_2$SO$_4$ aqueous solution (diluted from concentrated H$_2$SO$_4$ using Mili-Q water), ethanol, and Mili-Q water sequentially with ultrasonication for 1 min each. Then it was dried in a vacuum at room temperature for electrochemical measurements. The process of pretreating Nafion N117 proton exchange membrane beginned with cleaning the membrane by soaking it in Mili-Q water for 1–2 h to remove contaminants, followed by thorough rinsing with fresh Mili-Q water. Next, the membrane underwent acid treatment by immersing it in 0.5 M sulfuric acid solution for 1–2 h at 80 °C. After acid treatment, it is imperative to rinse the membrane thoroughly with Mili-Q water to remove any residual acid. Then, the membrane was stored in Mili-Q water to prevent it from drying out. A piece of Nafion N117 membrane (1 × 1 cm$^2$) was tailored for the H-cell. CV measurements were carried out on a 1 × 1 cm$^2$ piece of bulk Ni foam in 0.1 M NaOH using a typical H-type cell, with 20 mL of electrolyte placed in each chamber. An Ag/AgCl reference electrode filled with saturated KCl was employed as the reference electrode (RE), while a graphite rod with a large surface area served as the counter electrode (CE). The conversion from Ag/AgCl (sat. KCl) to the RHE scale in 0.1 M NaOH was determined from the Nernst relationship, giving a potential difference of 0.9666 V. All electrochemical data were recorded and analyzed using EC-Lab software.

## Data availability
The Source data underlying the figures of this study are available with the paper. All raw data generated during the current study are available from the corresponding authors upon request. Source data are provided with this paper.

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

## Acknowledgements

C.W., R.R., B.L.F., W.C., B.A.H., K.A., and Q.Z. acknowledge the support from the Center for High Precision Patterning Science (CHiPPS), an Energy Frontier Research Center program funded by the U.S. Department of Energy, Office of Science, Office of Basic Energy Sciences. J.Y. and H.L. acknowledge the support from the Liquid Sunlight Alliance (LiSA), which is supported by the U.S. Department of Energy, Office of Science, Office of Basic Energy Sciences, Fuels from Sunlight Hub under Award No. DE-SC0021266. B.A.H. and Z.P. acknowledge the support from the U.S. Department of Energy, Office of Science, Office of Basic Energy Sciences, Materials Sciences and Engineering Division under contract No. DE-AC02-05CH11231, Unlocking Chemical Circularity in Recycling by Controlling Polymer Reactivity across Scales program CUP-LBL-Helms. This research used resources at the Advanced Light Source of Lawrence Berkeley National Laboratory, supported by U.S. Department of Energy, Office of Science, Office of Basic Energy Sciences under Contract No. DE-AC02-05CH11231. This work used nanofabrication facilities at the Molecular Foundry of Lawrence Berkeley National Laboratory, supported by U.S. Department of Energy, Office of Science, Office of Basic Energy Sciences under Contract No. DE-AC02-05CH11231. The authors acknowledge the National Energy Research Scientific Computing Center (NERSC) provide supercomputer resources to this work for the FEM-based simulations. The authors also acknowledge Dr. Thomas Ferron, Dr. Bernhard Luttgenau and Dr. Zachary Fink from Advanced Light Source of Lawrence Berkeley National Laboratory for their support of the manuscript revision and Dr. Yue Liu from Chemical Sciences Division of Lawrence Berkeley National Laboratory for her support of the characterization of the Ni LGNPs. Y.Y. was partially supported by the Center for Alkaline-Based Energy Solutions (CABES), an Energy Frontier Research Center program supported by the U.S. Department of Energy, under grant No. DE-SC0019445, and was partially supported by the Cornell Atkinson Center for Sustainability and the Kavli Institute at Cornell (KIC) Instrumentation Grant. Y.Y. also acknowledges the support from the Berkeley Miller Research Fellowship.

## Author contributions

H.L., K.A., Q.Z., and I.A.C. contributed equally to this work. C.W. and H.L. supervised this work. I.A.C. and C.W. originated the idea of this work. H.L. led the experimental efforts and manuscript writing. K.A. and G.F. conducted scattering modeling and FEM-based simulations. I.A.C., Z.P. designed and fabricated in-situ cells. H.L. I.A.C., Z.P., and F.Y. carried out *operando* PE-RSoXS experiments and Y.Y. and H.L. performed the electrochemical measurements. Q.Z., S.D., and W.C. conducted the nanofabrication of Ni LGNPs. H.L., K.A., Q.Z., Z.P., Y.Y., and C.W. performed the data analysis. H.L., K.A., Q.Z., I.A.C., Y.Y., Z.P., A.H., B.A.H., B.L.F., R.R., J.G., W.Y., J.Y., and C.W. co-wrote the manuscript. All authors contributed to the discussion on data interpretation and approved the submission of the manuscript.

## Competing interests

The authors declare no competing interests.
