## [Transparent Peer Review file · Nature Communications]

Pattern-enhanced Resonant Soft X-ray Scattering for Operando Monitoring of Electrochemical Solid-Liquid Interfaces

Corresponding Author: Dr Cheng Wang

Version 0:

Reviewer comments:

Reviewer #1

(Remarks to the Author)

In this manuscript, Li et al. present the feasibility of the PE-RSoXS measurement technique using nickel linear grating nanopattern electrodes; however, the study falls short of establishing the significance of this approach for the broader electrocatalysis field. The reported findings, namely oxidation state changes and oxide layer dynamics, largely confirm well-established phenomena without offering novel insights or advancing the current understanding of electrode-electrolyte interfaces. Furthermore, the requirement for specialized nanopatterned electrodes significantly limits the method's applicability to practical electrocatalytic systems, where powder-based catalysts dominate. These critical limitations suggest that the manuscript is unlikely to attract broad interest and therefore does not align with the scope of Nature Communications. Instead, it may be more appropriate for publication in a specialized journal focused on advanced characterization techniques. The main concerns are as follows:

1. In line 33, The authors claim that PE-RSoXS enables high spatial and temporal resolution characterization of solid-liquid interfaces appears overstated based on the presented evidence. The technique was primarily employed to monitor oxidation state changes in Ni LGNPs during OER, information that could be similarly obtained through conventional soft XAS. Furthermore, the simulation-derived estimates of nanoscale oxide layer thickness variations do not seem to offer substantial advantages over established electron microscopy techniques, either in terms of resolution or mechanistic insight.
2. In line 69, While the authors assert that PE-RSoXS offers unprecedented spatial resolution and chemical sensitivity, the presented results fail to demonstrate any substantial improvement over existing characterization techniques. The observation of Ni oxidation to NiOOH during OER is a well-established phenomenon, which does not sufficiently showcase the unique advantages of this technique. Therefore, the authors are encouraged to further clarify what new or distinctive insights PE-RSoXS reveals beyond what is already accessible through conventional methods, in order to highlight the necessity and uniqueness of this approach.
3. In line 149, The authors claim that the exposure time of PE-RSoXS can reach the microsecond scale; however, no dynamic processes matching this timescale were observed in the current study. The authors are therefore encouraged to investigate processes whose dynamics better align with this temporal resolution, in order to obtain more compelling and meaningful results.
4. In line 254, the authors use simulations to reveal the potential-induced change in oxidation layer thickness of the Ni-based catalyst. However, there is a lack of direct experimental evidence to support this claim. Since the oxidation of Ni during the OER is generally irreversible, changes in the oxide layer thickness should be observable by electron microscopy. Therefore, additional experimental evidence is needed to support the PE-RSoXS observations.
5. The authors provide a detailed discussion of PE-RSoXS principles and technical merits in the main text. While this information is valuable, moving it to the supplementary information would improve the manuscript's focus and clarity. The main text should instead provide a concise and clear discussion that highlights the distinctive structural features revealed by PE-RSoXS and emphasizes the necessity of this technique for the study. This approach will attract greater interest from readers.
6. The PE-RSoXS measurements require highly specialized sample preparation to produce well-ordered Ni line-grating nanopatterns. Although this approach is interesting, its broader applicability, particularly to conventional powder-form nanocatalysts, remains to be demonstrated.

Reviewer #2

(Remarks to the Author)

The manuscript describes an implementation of the emerging RSoXS scattering method that exploits a Bragg structure sample to gain strong sensitivity to the composition within it. The new implementation is coined “PE-RSoXS.” It is demonstrated in a Nickel electrochemical oxidation experiment, and different optical constants are pulled out that describe different shell layers obtained at different electrochemical potentials. The analysis framework features a full Maxwell simulation beyond typical Born approximation approaches.

The manuscript reflects important advances and should be considered for publication in Nature Communications, provided the issues identified in review are addressed. Although this reviewer will provide extensive feedback for the author team (see below), the volume of feedback does not reflect manuscript quality. There are issues in presentation and context, and some technical issues that I will highlight in detail. Provided the editors agree, I look forward to continued consideration of this article for publication in this fine journal.

- Overarchingly, the centerpiece of the manuscript describes a technique that is usually called “diffraction anomalous fine structure,” or DAFS. This technique is actually very old. It has been described, in some form, since 1956 - Y. Cauchois (1956) — “Distribution spectrale observée dans une région d’absorption propre de divers cristaux”, C. R. Acad. Sci. Paris 242, 100–102. The idea was extended in work by other authors in the mid-70s, and re-emerged as a method named DAFS in the surface science era of the early-to-mid-90s, with a flagship publication led by Joe Woicik who is now at NIST: Stragier, H., Cross, J. O., Rehr, J. J., Sorensen, L. B., Bouldin, C. E., & Woicik, J. C. (1992). Diffraction anomalous fine structure: A new x-ray structural technique. Phys. Rev. Lett., 69, 3064–3067. doi:10.1103/PhysRevLett.69.3064.

None of this rich history is acknowledged in the manuscript under review, nor are there any explanations of how the current manuscript goes beyond. In this reviewer’s eyes, there are at least two claims of novelty / advance for the manuscript under review: 1) there are very few implementation of DAFS with soft X-rays on larger Bragg structures, and 2) the simulations in the manuscript under review and the method of obtaining optical constants are extremely sophisticated compared to most previous DAFS examples, and particularly well-suited to soft X-rays given their high likelihood of multiple scattering. The authors should acknowledge the rich history of DAFS (a nice 2018 review is here <https://arxiv.org/pdf/cond-mat/0309624> but I don’t know if it was published outside arxiv), and then explain their advances.

- The reviewers make multiple claims of interface sensitivity - the word is mentioned 45 times in the article! But there is nothing especially interface sensitive about RSoXS or PE-RSoXS. The RSoXS technique is FORM FACTOR sensitive. When the form factor includes an interface, we can of course be sensitive to it, and different placements of the interface within the form factor (and within the structure factor it is convoluted with) will result in different levels of sensitivity.

Yes, in the implementation here, the authors demonstrate sensitivity to a shell in a core-shell structure (this is the right language, not “interface sensitive”), but there will also be sensitivity to other parts of the structure that are not interfacial. The technique will be excruciatingly sensitive to the density and composition of the solid core, for example.

The reason this reviewer is so adamant about the language used here is that the “interface sensitive” claims are misleading and they echo misleading claims made about RSoXS about 10-15 years ago that have (in this reviewer’s opinion) sown confusion and stunted the growth of the technique. Much early work on the RSoXS of organic electronics and organic photovoltaics (OPVs) in particular made claims about special sensitivity to interfacial composition and orientation. The language used then (and now, in the article under review!) was extremely similar to the language of nonlinear optics techniques such as sum frequency generation (SFG) (which was emerging and highly visible at the time), in which symmetry breaking at interfaces truly affects spectroscopic selection rules, allowing spectra to be observed that isolate signals from the interface. That is NOT what is happening in RSoXS – scattering sees everything, not just interfaces, and RSoXS is frequently bulk-dominated. But this article strongly implies some kind of special sensitivity to interfaces with the 45 mentions of the word “interface” and in fact these authors use “symmetry breaking” language to explain their results in lines 127-128, when what is meant is that subtle perturbations in the form factor can have strong effects on order expression. But these perturbations don’t necessarily have to be at interfaces, and RSoXS is not sensitive to symmetry breaking specifically, just the form factor. The authors should remove their misleading language about interface sensitivity and symmetry breaking. If they would like to include it, they should include the context that the technique is sensitive to interfaces “when you know there is an interface, and you know where it is, and you can model it”. That is what is required. It’s also the same way the RSoXS is sensitive to aspects of structure other than interfaces.

- The above concern brings me to an adjacent point, which is that the authors have not demonstrated the “spatial resolution” that they mention 19 times in the manuscript. What they have demonstrated is fine sensitivity to shell thickness in a core-shell structure. Which, in this reviewer’s opinion, should certainly be enough for publication in Nat Comm if the authors are willing to soften their claims. But “spatial resolution” implies that they can identify “where” that material is within the structure. Instead, the authors have assumed that it is on the sidewall of the solid structure. That is, of course, a perfectly valid assumption given the electrochemical nature of the experiment, but it does not demonstrate spatial resolution.

To demonstrate spatial resolution, the authors could consider simulating what happens if the shell were “free floating” within the electrolyte-filled part of the Bragg structure, spaced by various distances from the wall. As unphysical as those models would be, that could potentially demonstrate spatial resolution. I can assert from personal experience that scattering from Bragg structures is typically far less sensitive to specific placement within the Bragg structure (“spatial resolution”) than it is sensitive to the specific volume fractions (shell thickness) and their indices (shell and core and matrix composition), but I would defer to simulation results from the authors if they showed otherwise.

- There is a significant missed opportunity to structure the narrative here with deeper fundamentals in the presentation of the “scattering derived spectra.” These appear to be [I integrated across a peak] vs. energy, presented in Fig 2 d-e-f. On lines 190-191 these spectra are directly compared to X-ray absorption spectra data, and described as “comparable.” But it is incorrect to compare them because [I integrated across a peak] is the result of *differences between indices* of the volumes within the scattering structure, and it includes the delta parts of those indices. So, there is dispersion baked into that “spectra”, and even if you could isolate beta, it would be a difference spectrum between nickel beta and nickel oxide beta (electrolyte beta would be reasonably flat, but sloped). The authors should remove this comparison to XAS, and more properly acknowledge the contributions to I vs. energy.

But there is an even stronger way to do that, and that is to lean into the concept of the total scattering invariant (TSI). The TSI describes the integrated scattering across all q and in all 3 dimensions; for RSoXS patterns an approximation frequently called “integrated scattering intensity” (ISI) is used by integrating $I \cdot q^2$ (sometimes q-bounded or feature-bounded, and sometimes for all measurable q) vs energy. The authors could consider presenting Fig 2 d-e-f as feature-bounded ISI vs. energy. The reason this would be valuable is that, in principle, TSI / ISI should be directly proportional to the sum of the combinatorial pairwise binary contrasts ($\Delta(\beta)^2 + \Delta(\delta)^2$) weighted by their volume fraction products (see Collins and Gann 2021 Eq 8 and Eq 15; it can be extended to any number of components). Quantitative agreement of the energy-dependent ISI vs. energy and expectation contrast was robustly demonstrated in Ferron, Pope, and Collins PRL 2017, and it underpins the physics of RSoXS and of the manuscript under review. The Maxwell solver will reproduce this physics but the authors should more robustly acknowledge it, as they deal with I vs. energy much earlier than they introduce n, delta, beta, etc, when they arise from a common framework.

The authors’ mentions of “numbers of scatterers N” on lines 155-156 and the sensitivity to N squared are a confusing and less general recapitulation of the physics I mention above. They are misleading in stating that “conventional scattering” has linear sensitivity to the number of scatterers. All SAS has $\Delta(\rho)^2$ magnitude; the number of scattering objects increases the scattering intensity linearly in the dilute form factor limit, whereas in a Bragg structure you have a *really* strong reporter of $\Delta(\rho)^2$. Perhaps that’s what the authors intended to say but the concepts are muddled here.

- The authors should moderate their language and results that suggest spatial resolution beyond the diffraction limit, and they need to be clear about their interaction with this limit. Spatial resolution beyond the diffraction limit is implied by their language: “atomic scale resolution” is used line 73. But the wavelength here would be ~1.5 nm, with a diffraction limit of 0.75 nm. The PE-RSoXS technique – like every other SAS technique that exists - should not be capable of real spatial resolution beyond that diffraction limit. But, the authors do demonstrate major differences in the I vs. energy simulation for 0.2 nm increments in shell thickness. This does NOT demonstrate 0.2 nm spatial resolution, it goes back to my earlier point that we are exquisitely sensitive to volume fraction of the shell (or an alternative way of thinking is the effective medium index within diffraction limit volumes), and less sensitive to where the shell is placed within the structure (what we would call “spatial resolution”). If we assume or have robust proof that the shell must be attached to the sidewall, then we have an ~ersatz sub-diffraction-limit spatial resolution, but the assumption about the location is doing all the heavy lifting in that result. I strongly recommend the authors reconsider their language and directly address their interaction with the diffraction limit lest they mislead readers.

- Aside from my points above, the introduction is otherwise very accessible and explains some key jargon pieces such as the meaning of “near field.” However, the description of the modeling – a centerpiece of the manuscript – is extremely confusing. Some questions that came up during my reading:

. what is the “energy offset”? I’m guessing it’s to accommodate differences between the energy of the beamline at RSoXS measurement and the energies of the NEXAFS-derived library index used for the core. But this is never explicitly stated in the body or methods.

. was the shell thickness floated in the fit? This is never explicitly stated.

. I’m quite surprised at how nicely smooth the delta and beta DAFS extractions are. Did the authors use tricks to keep them smooth – like a thickness constraint or the use of the previous energy’s index as an initial guess for the next? These kinds of fittings are prone to local minima.

- The authors should acknowledge some limitations around model uniqueness. Importantly, the simulations are done on structures that perfect, and the real structures are not. An imperfection that strongly affects order expression (the principal thing that is fit here) is interfacial width / roughness, which is strongly convoluted with sidewall angle in q_{xy} scattering. The authors assume that the electrochemically converted material and its two interfaces are sub-atomically smooth. Any line-edge roughness or shell thickness inhomogeneity (and there is usually a significant amount with solid-state electrochemistry) in the core or shell will diminish higher peak orders relative to lower ones and potentially could be misfit. I’m not suggesting the authors need more parameters for their model, but they should acknowledge this limitation and discuss its potential effects on the results.

- The authors perhaps overemphasize odd/even effects; these are not universal in Bragg structures and are only strongly relevant near the 50% duty cycle limit. I bring this up because it oversimplifies the physics - different fill ratios will produce different harmonic effects on order expression: 33% / 66 % will suppress every 3rd order, etc. I leave it to the authors to consider this point.

(Remarks to the Author)

This manuscript reports the use of pattern-enhanced resonant soft X-ray scattering (PE-RSoXS) as a new probe of chemical and structural dynamics at electrochemical solid–liquid interfaces. The methodology development is technically impressive. I recommend publication after addressing the following concerns:

1. “In situ”, instead of “operando” is more accurate to describe the present methodology.
2. Atomic-scale resolution might be overstated while the results mainly reach sub-nanometer sensitivity.
3. A comparison table summarizing state-of-the-art interface characterization techniques (e.g., TEM, XAS, AP-XPS, fluorescence microscopy, RSoXS) would greatly strengthen the introduction and highlight the advantages of PE-RSoXS.
4. Please comment on the generality of the technique. Could this approach be applied to other catalytic materials (e.g., Co or Cu) that undergo significant restructuring?
5. The method is described as “non-destructive,” but more evidence should be provided. While 1 ms exposure per energy point is reported, is this duration sufficient to fully avoid beam-induced damage? Is it a trade-off or the minimum exposure time? Would longer exposure time improve data quality and accuracy to assess the simulation?
6. The experimental and simulated Ni L-edge spectra do not fully coincide (e.g., 3rd-order in Fig. 2d). Please clarify the origin of these discrepancies. The spectral difference for experimental vs. simulated spectra should be plotted.
7. In Table 1, the shell thickness increases with potential, as expected from hydroxide intercalation. However, the core width is supposed to decrease as well. Could you explain this discrepancy?
8. Is there any conditioning procedure before the electrochemical measurement? If so, would the conditioning alter the core-shell restructuring? Beyond the OCP/1.2 V/1.6 V comparison, is it possible to assess the transition from pristine Ni (air/water-exposed) to OCP?
9. Is there any potential bubble issue, due to the beam-induced water radiolysis or generated O₂ bubble at 1.6V? Would it alter the electrolyte thickness and affect the scattering interpretation?
10. Could you provide the SEM/AFM analysis of the nanopatterns after experiments to confirm the structural integrity.

Version 1:

Reviewer comments:

Reviewer #1

(Remarks to the Author)

The authors have addressed the major concerns raised in the previous review round, and the revised manuscript now meets the high standards required for publication in Nature Communications. Nevertheless, several points could be further improved:

1. The distinction between “Operando” and “In situ” measurements should be used more rigorously. As the present experiments probe structural and chemical evolution under controlled electrochemical conditions without directly correlating these changes to real-time catalytic performance, the term “In situ” appears more appropriate.
2. While the manuscript correctly attributes the detection of ~0.2 nm structural changes to diffraction-intensity sensitivity rather than direct imaging beyond the diffraction limit (~0.75 nm), phrases such as “sub-nanometer resolution” or “visualization at sub-nanometer scales” may still be misleading. Consistent use of terms like “sub-nanometer sensitivity” would improve clarity.
3. The generality and limitations of the PE-RSoXS method would benefit from clearer discussion. In particular, it would be useful to clarify whether the approach fundamentally relies on highly periodic nanostructures and to what extent it can be extended to non-periodic, multiscale, or porous electrocatalyst systems relevant to practical applications.

Reviewer #2

(Remarks to the Author)

I am satisfied that the authors not only agree with me on almost every point, but they have fully addressed my concerns with important and substantial revisions.

We can quibble about whether PE-RSoXS is really a “new class” of measurement or just an evolution of DAFS with an artificial vs. naturally-occurring Bragg structure, but the current article provides sufficient context to let the reader decide.

Reviewer #3

(Remarks to the Author)

The concerns have been addressed.

Reviewer: 1

In this manuscript, Li et al. present the feasibility of the PE-RSoXS measurement technique using nickel linear grating nano-pattern electrodes; however, the study falls short of establishing the significance of this approach for the broader electrocatalysis field. The reported findings, namely oxidation state changes and oxide layer dynamics, largely confirm well-established phenomena without offering novel insights or advancing the current understanding of electrode-electrolyte interfaces. Furthermore, the requirement for specialized nanopatterned electrodes significantly limits the method's applicability to practical electrocatalytic systems, where powder-based catalysts dominate. These critical limitations suggest that the manuscript is unlikely to attract broad interest and therefore does not align with the scope of Nature Communications. Instead, it may be more appropriate for publication in a specialized journal focused on advanced characterization techniques. The main concerns are as follows:

Response: We sincerely appreciate the Reviewer's evaluation of our work, which has been invaluable in refining the manuscript to appeal to a broader audience across diverse fields. Here we agree that powder-based catalysts remain the dominant form in practical electrocatalysis, offering clear advantages for large-scale synthesis and deployment [Seh et al. *Science* 355, eaad4998 (2017); Zou et al. *Chem. Soc. Rev.* 44, 5148–5180 (2015)]. Resonant soft X-ray scattering (RSoXS) has been indeed successfully applied to powder electrocatalysts in recent high-impact studies [Yang et al. *J. Am. Chem. Soc.* 144, 8927–8931 (2022); Yang et al. *Nature* 614, 262–269 (2023); Yang et al. *Nat. Catal.* 8, 579–594 (2025)] with broad recognition, demonstrating its strong capability to *operando* probe chemical and structural dynamics of active species in disordered systems during electrocatalysis. However, a persistent challenge in applying RSoXS to directly probe complex electrochemical solid-liquid interfaces lies in quantitatively isolating weak and buried interfacial signals from bulk contributions when the system lacks well-defined geometry.

To address this, we deliberately developed a Ni line-grating nanopattern (Ni LGNP) electrode as a model platform for RSoXS to achieve methodology innovation in X-ray spectroscopy and scattering. This “sample-as-optics” approach transforms the electrode into an active optical element, enabling Bragg diffraction enhancement, near-field optical modulation, and diffraction-order-dependent separation of interfacial signals from bulk ones in patterned-enhanced RSoXS (PE-RSoXS), which have not been achievable with randomly oriented powders. This controlled architecture allows us to quantitatively benchmark the technique's sub-nanometer spatial resolution, chemical sensitivity, and *operando* low-dose performance before extending it to more complex powder systems. While powders offer potentials for practical applications, **our work aims to demonstrate that nanopatterns uniquely serve as an optical testbed** that amplifies weak buried interfacial signals and ensures statistical representativeness, thereby establishing the fundamental principles and performance limits of PE-RSoXS for future adaptation to industrially relevant electrocatalysts.

Compared with other conventional *operando* techniques, PE-RSoXS offers a unique combination of element-specific sensitivity, statistical representativeness, sub-nanometer dimensional sensitivity, and millisecond temporal resolution under realistic electrochemical conditions, while operating at ultra-low X-ray doses ($< 0.01 \text{ mJ}\cdot\text{cm}^{-2}$). For example, while *operando* transmission electron microscopy (TEM) enables real-time imaging, high-energy electrons enable damage on sensitive samples and often probe only localized regions. Synchrotron-based surface-sensitive methods such as X-ray absorption spectroscopy (XAS), ambient-pressure X-ray photoelectron spectroscopy (APXPS), and X-ray fluorescence microscopy (XFM) provide robust averaged chemical and structural information but generally lack the high spatial and/or temporal resolution required to resolve reaction fronts at the (sub-)nanometer and (sub-)millisecond scales. PE-RSoXS uniquely allows us to detect oxidation-state gradients, sub-nanometer swelling/contraction, and redox-coupled

morphological changes at buried solid-liquid interfaces in real time, which are extremely challenging to resolve simultaneously with existing probes. PE-RSoXS achieves high efficiency by using engineered periodic nanostructures to amplify weak resonant scattering signals, allowing rapid and statistically robust *operando* characterization compared to conventional methods.

To assess the feasibility and sensitivity of a new methodology, it is often effective to begin with a well-established phenomenon, which is a widely accepted scientific strategy [Collins et al. *Nat. Mater.* 11, 536–543 (2012); Kolmakov et al. *Nat. Nanotechnol.* 6, 651–657 (2011); Thibault et al. *Nature* 494, 68–71 (2013); Feijóo et al. *J. Am. Chem. Soc.* 145, 20208–20213 (2023)]. The oxidation-state transitions and oxide-layer dynamics observed in this work are consistent with the established oxygen evolution reaction mechanism on Ni-based electrocatalysts. The ability to quantitatively resolve these processes with high spatial and temporal resolution, as well as chemical precision represents a methodological advance that enables new classes of mechanistic studies. Furthermore, mechanistic insights gained from such precisely patterned electrodes, including the coupling between oxidation-state evolution, sub-nanometer structural changes, and electrocatalytic function, can directly inform the rational design of practical powder-based electrocatalysts. In this way, the nanopatterned electrode serves as a well-controlled testbed for uncovering generalizable interfacial principles that are directly translatable to industrial catalyst systems [Lum et al. *Energy Environ. Sci.* 11, 2935–2944 (2018); Liu et al. *ACS Appl. Mater. Interfaces* 13, 40513–40521 (2021)].

In summary, while the present study employs a model nanopatterned electrode to establish and validate PE-RSoXS, the broader significance lies in the development of a new and interface-specific *operando* X-ray methodology that overcomes longstanding barriers in characterizations of electrochemical interfaces. With its unique combination of signal amplification, diffraction-order selectivity, and element-specific *operando* capability, we believe PE-RSoXS will be of strong interests to the wider electrocatalysis, materials science, and interfacial chemistry communities, aligning well with the multidisciplinary scope of *Nature Communications*.

We have addressed each of the Reviewer's detailed comments point-by-point below. We hope that with these revisions, the manuscript will be considered suitable for publication in *Nature Communications*.

1. In line 33, The authors claim that PE-RSoXS enables high spatial and temporal resolution characterization of solid-liquid interfaces appears overstated based on the presented evidence. The technique was primarily employed to monitor oxidation state changes in Ni LGNPs during OER, information that could be similarly obtained through conventional soft XAS. Furthermore, the simulation-derived estimates of nanoscale oxide layer thickness variations do not seem to offer substantial advantages over established electron microscopy techniques, either in terms of resolution or mechanistic insight.

Response: We thank the Reviewer for raising these important points, which allow us to clarify the unique capabilities of PE-RSoXS. Firstly, the high spatial and temporal resolution of PE-RSoXS demonstrated here is directly supported by our experimental and simulation results. We resolved sub-nanometer structural dynamics (shell thickness changes of Ni LGNPs) directly at electrochemical solid-liquid interfaces based on our experimental and simulation results (**Table 1** in the revised manuscript), while simultaneously distinguishing oxidation-state gradients between the metallic Ni core (from 105.6 nm to 107.7 nm) and oxidized Ni shell (from 4.1 nm to 6.4 nm) through diffraction-order-dependent spectral separation. Temporally, PE-RSoXS achieves single-shot diffraction pattern acquisition with 1 ms exposure time, enabling *operando* monitoring of dynamic processes with ultra-low X-ray doses ($< 0.01 \text{ mJ}\cdot\text{cm}^{-2}$) that avoid beam-induced chemical changes. This combination of sub-nanometer dimensional sensitivity,

millisecond temporal resolution, and element-specific *operando* capability under realistic electrochemical conditions is not achievable with conventional soft XAS or electron microscopy and is central to the methodological advance established in this work.

Here we summarize the distinction of PE-RSoXS from other conventional techniques detailly in **Table R1**. We agree with reviewer that oxidation-state and structural changes in Ni LGNPs during OER could, in principle, be detected by conventional soft XAS and electron microscopy techniques respectively.

Table R1. Comparison of Characterization Techniques for Electrochemical Interfaces.

Technique	PE-RSoXS	TEM	XAS	APXPS	XFM
Dimensional Sensitivity	Sub-nanometer (via diffraction-order sensitivity and near-field optical modulation)	Atomic-scale (< 0.1 nm) imaging possible	Tens to hundreds of nanometers (limited by beam spot size and detection geometry)	Few nanometers sampling depths in solids; lateral resolution: nanometers to micrometers (limited by beam spot size and detection geometry)	Tens to hundreds of nanometers (microprobe); < 10 nm with ptychography
Temporal Resolution	Millisecond scale (single-shot diffraction patterns; compatible with ultrafast detectors)	Millisecond to minutes (with fast cameras)	Seconds to minutes per spectrum (energy scans)	Seconds to minutes; milliseconds possible with fast detectors	Seconds to minutes per scan
Chemical Sensitivity	High (tunable to specific absorption edges; diffraction-order selectivity isolates interfacial species)	Limited (requires EELS or EDS)	High (direct absorption edge tuning)	High (chemical specificity; oxidation states and bonding)	High (Element-specific via fluorescence yield)

Interface Accessibility	High (coherent amplification of weak buried interfacial signals)	Limited (requires sample thinning or cross-section preparation ; interfaces may be altered during the sample preparation)	Moderate (surface-sensitive but buried interfacial signals diluted by the bulk signals)	Moderate (Surface and sub-surface accessible; buried solid-liquid interfaces challenging, limited by probing depth)	High (Bulk and buried interfaces accessible)
Environment Compatibility	Fully compatible with realistic liquid cells and aqueous conditions	Fully compatible with realistic liquid cells and aqueous conditions	Fully compatible with realistic liquid cells and aqueous conditions	Compatible with near-ambient pressure gases; liquids via microjets or meniscus methods	Compatible with air, vacuum, and liquid cells
Radiation Dose / Damage Risk	Low (can be lower than $0.01 \text{ mJ}\cdot\text{cm}^{-2}$ per frame)	High (beam-induced structural and chemical changes, especially in soft materials)	Moderate dose; soft X-rays more susceptible to chemical changes	Moderate dose; soft X-rays more susceptible to chemical changes	Generally low dose per pixel; scanning can accumulate dose
Statistical measurement	High (averages over hundreds of identical nanopattern units depending on the beam spot size)	Limited (small field of view, typically single particles or local regions)	High (averages over large illuminated volume)	Moderate (can measure multiple surface regions but limited penetration depth)	High (can map statistical areas for a large range)

Measurement Efficiency	High (diffraction encodes structural order while resonance tuning provides chemical-state specificity concurrently)	Low (sequential detections with structural and chemical information with EELS/EDS)	Low (primarily provides chemical-state information ; structural evolution must be measured separately with other techniques)	Low (primarily provides chemical-state information; structural evolution must be measured separately with other techniques)	Low (primarily elemental mapping, structural information indirect)
Key Strengths	Interface-specific operando probing of structural and chemical dynamics; low-dose; diffraction-order selectivity	Direct atomic-scale imaging and crystalline structures	Chemical-state analysis of surfaces and sub-surface regions	Direct surface and sub-surface chemical state under near-realistic conditions	High sensitivity to trace elements; quantitative elemental mapping
Key Limitations	Requires patterned structures for coherent enhancement	Beam-induced changes; limited detection area; difficult to operando probe buried interfaces	Limited spatial resolution; bulk-interface separation challenging	Limited penetration depth; challenging for fully buried interfaces	Limited chemical state information; slow scan for high-resolution maps

We also highlight that PE-RSoXS can provide several critical advantages that go beyond the conventional methods as follow.

(1) Buried Interface Sensitivity and Bulk/Interface Signal Separation

PE-RSoXS employs a “sample-as-optics” design in which engineered periodic nanopatterns coherently amplify weak individual interfacial scattering signals. Spatial chemical mapping of each unit by diffraction intensity analysis allows us to quantitatively decouple buried interfacial contributions from bulk signals. But conventional soft XAS cannot achieve it because absorption spectra inherently mix bulk and interface responses without dimensional sensitivity.

This capability is particularly important for electrochemical systems where buried interfaces govern catalytic activity but are difficult to isolate spectroscopically.

(2) *Operando* Probing Structural and Chemical Dynamics Simultaneously

Unlike conventional soft XAS, which primarily provides chemical-state information, PE-RSoXS simultaneously resolves sub-nanometer structural dynamics (e.g., swelling/contraction of line-grating features via diffraction-order intensity changes) and chemical-state evolution (via energy-dependent scattering spectra) in a single *operando* measurement. This dual capability allows direct correlation between oxidation-state transitions and morphological changes at buried solid-liquid interfaces with sub-nanometer precision and millisecond temporal resolution, enabling mechanistic insights into structure–function relationships that are inaccessible when using conventional soft XAS alone [Magnussen et al. *Chem. Rev.* 124, 629–721 (2024); Chen et al. *Chem. Rev.* 124, 5421–5469 (2024); Timoshenko et al. *Chem. Rev.* 121, 882–961 (2021)]. While TEM can achieve high-resolution imaging at atomic scales, it's difficult to probe buried interfaces in liquid environments without invasive sample preparation, nor can it easily provide element-specific chemical-state information without additional techniques such as EELS, which often compromise temporal resolution and increase beam damage. In contrast, PE-RSoXS achieves sub-nanometer dimensional sensitivity to buried interfacial structural variations via coherent diffraction enhancement, while simultaneously *operando* delivering element-specific chemical information in liquid cells. The simulation-derived shell thickness variations are extracted from multi-order diffraction spectra, representing statistical and quantitative interfacial measurements under realistic electrocatalytic conditions, where TEM's applicability is highly constrained [Chao et al. *Chem. Rev.* 123, 8347–8394 (2023); Li et al. *Chem. Rev.* 123, 10728–10749 (2023)].

(3) Statistical Representativeness under *Operando* and Low-Dose Conditions

PE-RSoXS measurements average over hundreds of identical nanopattern units within the beam field, ensuring statistical reliability while operating at ultra-low X-ray doses ($< 0.01 \text{ mJ}\cdot\text{cm}^{-2}$). This largely avoids beam-induced chemical changes and allows long-term *operando* monitoring in liquid electrochemical cells. However, TEM suffers from severe beam damage and limited representativeness due to its inherently localized imaging [Chen et al. *Adv. Mater.* 32, 1907619 (2020); Nevison-Andrews et al. *Chem. Phys. Rev.* 6, 031302 (2025)].

(4) Mechanistic Insights over Ni LGNPs

The Ni LGNP system was chosen as a controlled model platform to rigorously validate the PE-RSoXS methodology, not to claim new chemistry in OER on Ni-based electrocatalysts. By demonstrating the ability to correlate oxidation-state gradients and sub-nanometer dimensional changes in real time, this work establishes PE-RSoXS as a generalizable approach for studying dynamic solid-liquid interfaces in electrocatalysis, energy storage, and other interfacial systems. Insights from such controlled studies can directly inform the design of practical powder-based catalysts, where similar phenomena occur but are harder to measure via conventional probes.

These capabilities, demonstrated here for the first time in an *operando* electrochemical system, represent the central methodological advance of PE-RSoXS and distinguish it from both conventional soft XAS and electron microscopy techniques.

Revision made: We added **Table R1** as **Supplementary Table 1** and the following illustration into the revised Supplementary Information (SI) as **Supplementary Note 1** to make a comprehensive comparison of PE-RSoXS with other conventional techniques for probing electrochemical interfaces with relevant references to support our demonstration.

“Compared to other state-of-the-art interface characterization methods such as transmission electron microscopy (TEM), X-ray absorption spectroscopy (XAS), ambient-pressure X-ray photoelectron spectroscopy (APXPS), and X-ray fluorescence microscopy (XFM), Pattern-Enhanced Soft X-ray Scattering (PE-RSoXS) offers a unique combination of high statistical representativeness, sub-nanometer dimensional sensitivity, and element-specific chemical-state resolution, all achievable under realistic electrochemical operating conditions. Unlike TEM and APXPS, which are limited in probing fully buried solid-liquid interfaces, PE-RSoXS can access these environments without compromising spatial or chemical sensitivity. While XAS provides statistical chemical-state information, it lacks lateral resolution and diffraction-order selectivity for directly probing interfaces, which is inherent to PE-RSoXS through coherent scattering amplification from identical nanopatterns. Furthermore, the ultralow-dose capability of PE-RSoXS ($< 0.01 \text{ mJ}\cdot\text{cm}^{-2}$ per frame) minimizes radiation-induced artifacts, enabling *operando* monitoring of dynamic processes with millisecond temporal resolution, surpassing the typical time resolution of conventional methods. These combined attributes position PE-RSoXS as a powerful and complementary tool for probing chemical and structural dynamics at buried electrochemical solid-liquid interfaces. PE-RSoXS operates on principles analogous to protein crystallography, where diffraction from periodic units enables reconstruction of real-space features with atomic to nanometer-scale precision. In our case, the periodic nanopatterns define a structural unit cell, and the spatial distribution of chemical species within that unit cell modulates the scattering form factor in a position-sensitive manner. This coherence-based sensitivity to the distribution and geometry of resonant materials enables spatial resolution at sub-nanometer length scales, as supported by both the experimental data and the finite-element modeling.”

2. In line 69, While the authors assert that PE-RSoXS offers unprecedented spatial resolution and chemical sensitivity, the presented results fail to demonstrate any substantial improvement over existing characterization techniques. The observation of Ni oxidation to NiOOH during OER is a well-established phenomenon, which does not sufficiently showcase the unique advantages of this technique. Therefore, the authors are encouraged to further clarify what new or distinctive insights PE-RSoXS reveals beyond what is already accessible through conventional methods, in order to highlight the necessity and uniqueness of this approach.

Response: We appreciate the Reviewer’s request for the clarification on the distinctive insights provided by PE-RSoXS beyond what is accessible through conventional characterization methods. We agree that oxidation of Ni active sites during oxygen evolution reaction (OER) is a well-established phenomenon; this is precisely why we chose this model system to demonstrate the new *operando* X-ray methodology. We hope to clarify that the novelty of our work lies not in uncovering new chemistry of Ni-based electrocatalysts, but in establishing a new *operando* X-ray approach that can resolve both chemical-state evolution and sub-nanometer dimensional dynamics simultaneously at buried electrochemical solid-liquid interfaces.

Specifically, PE-RSoXS reveals several unique aspects that conventional techniques cannot achieve (**Table R1**).

(1) Simultaneous correlation of chemical and structural changes

By extracting energy-dependent scattering spectra from multiple diffraction orders, PE-RSoXS tracks oxidation-state transitions (Ni^{2+} to Ni^{3+}) while concurrently quantifying sub-nanometer dimensional variations of the nanopatterns (**Fig. 3d** and **Table 1** in the revised manuscript). This direct correlation between redox chemistry and morphological changes at buried interfaces is inaccessible to conventional soft XAS, which provides chemical-state information but lacks structural resolution; as well as to electron microscopy, which can image structures but requires separate, often beam-intensive methods for chemical mapping. By integrating nanopatterned electrodes as optical elements, PE-RSoXS boosts interfacial scattering intensity by orders of

magnitude, delivering simultaneous structural and chemical insights with sub-nanometer sensitivity, millisecond exposure, and low-dose measurements. This makes it a powerful and highly efficient tool for *operando* studies of complex interfaces.

(2) Diffraction-order-dependent separation of bulk and interfacial signals

The “sample-as-optics” nanopattern design enables Bragg diffraction enhancement and order-specific sensitivity, allowing us to decouple interface-specific signals (even orders) from bulk responses (odd orders) based on the 50% duty cycle structure of Ni LGNPs. This selective separation is not possible in conventional soft XAS, where absorption spectra inherently mix bulk and interface contributions; nor in TEM, where buried interfaces are difficult to probe *operando* in liquid cells.

(3) Buried interfacial sensitivity with statistical reliability under *operando* conditions

PE-RSoXS averages over hundreds of identical nanopattern units in the beam irradiation field statistically, ensuring robust sampling while operating at ultra-low X-ray doses ($< 0.01 \text{ mJ}\cdot\text{cm}^{-2}$) and millisecond exposure time per shot that avoid beam-induced chemical changes. This allows long-term *operando* monitoring of buried interfaces in realistic electrochemical environments, which is highly challenging for TEM due to beam damage and limited sampling coverage.

(4) Quantitative extraction of sub-nanometer shell thickness under realistic electrochemical conditions

The variations of the oxide-layer thickness derived from multi-order scattering spectra represent averaged and quantitative measurements of buried interfacial geometry under *operando* liquid-cell conditions. Conventional TEM can measure similar dimensions *ex situ*, but *operando* measurements in liquid are constrained by beam damage, small sampling areas, and the need for invasive preparation.

Thus, while the Ni transition itself is known, PE-RSoXS uniquely demonstrates the ability to quantitatively resolve and correlate chemical-state gradients and sub-nanometer structural dynamics at buried interfaces in real time, under realistic electrochemical conditions, with statistical reliability and minimal beam perturbation. Establishing these capabilities in a controlled model system is a necessary step toward applying the method to more complex and industrially relevant electrocatalysts and represents the core methodological advance of this work.

Moreover, we also provide several reported examples here demonstrating that using a known and well-established system to validate a new methodology is a standard and widely accepted scientific practice, particularly when the objective is to benchmark resolution, sensitivity, and *operando* capability. Collins et al. firstly demonstrated polarized soft X-ray scattering can probe molecular orientation down to size scales of 10 nm using well-known polymers [*Nat. Mater.* 11, 536–543 (2012)]; Feijóo et al. used CO₂ electroreduction on Cu nanoparticles as a typical platform to show the capability of high-energy-resolution fluorescence-detected XAS [*J. Am. Chem. Soc.* 145, 20208–20213 (2023)]; Chu et al. simply fabricated a stacked metal structure to present that multibeam scattering in grazing-incident reflection geometry is sensitive to 3D structures [*Nat. Commun.* 14, 5795 (2023)].

3. In line 149, The authors claim that the exposure time of PE-RSoXS can reach the microsecond scale; however, no dynamic processes matching this timescale were observed in the current study. The authors are therefore encouraged to investigate processes whose dynamics better align with this temporal resolution, in order to obtain more compelling and meaningful results.

Response: We thank the Reviewer for this valuable observation regarding the temporal resolution of PE-RSoXS. Indeed, while the current study achieved single-shot acquisition of

diffraction patterns with 1 ms of exposure time and demonstrated *operando* monitoring of electrochemical solid-liquid interfaces, we did not probe processes occurring on the microsecond timescale. Our statement that PE-RSoXS can reach the microsecond regime refers to the intrinsic capability of the technique and instrumentation. Specifically, its compatibility with high-brightness synchrotron sources, diffraction-limited storage rings, and ultrafast detectors (e.g., hybrid pixel array detectors), which can operate at frame rates exceeding 1 MHz. In such configurations, the amplification from nanopatterned structures enables sufficient signal-to-noise ratios even at the extremely short exposure time, which largely enhances the efficiency to *operando* probe interfacial information.

The focus of the present work was on establishing and validating PE-RSoXS as a new interface-specific *operando* X-ray methodology in a controlled model system, rather than to target the fastest possible dynamics. Ni LGNPs undergoing OER were chosen because their oxidation-state transitions and sub-nanometer dimensional variations are well-established and occur on timescales accessible in the millisecond range or slower ones, allowing us to benchmark dimensional and chemical sensitivity at the buried interfaces under realistic electrochemical conditions. This approach follows standard practice in methodology development, where known systems are used to validate core capabilities before applying them to more complex or faster processes.

In addition, the millisecond temporal resolution of PE-RSoXS is particularly valuable to avoid beam-induced chemical changes. The coherent Bragg diffraction enhancement from nanopatterned electrodes enables sufficient scattering signal to be captured in single-shot exposures as short as 1 ms, minimizing the total X-ray dose to below $0.01 \text{ mJ}\cdot\text{cm}^{-2}$ per frame. This ultralow dose is critical for avoiding beam-induced chemical or structural changes, which are well-documented in soft X-ray experiments on hydrated, organic, or nanomaterials [Wang et al. *J. Phys. Chem. B* 113, 1869–1876 (2009); Kubin et al. *Phys. Chem. Chem. Phys.* 20, 16817–16827 (2018)]. By acquiring high-quality diffraction and spectroscopic data before any significant radiation damage accumulates, PE-RSoXS preserves the native state of the solid-liquid interfaces throughout the measurements. This capability ensures that the observed oxidation-state transitions and sub-nanometer structural changes reflect the real behaviors of the electrocatalysts, rather than artifacts introduced by prolonged exposure. In this way, millisecond-scale PE-RSoXS not only enhances temporal resolution but also ensures the integrity of sensitive systems under realistic electrochemical conditions.

We agree that applying PE-RSoXS to phenomena whose dynamics match its full temporal potential would be compelling, such as electrochemical charging/discharging in microsecond regimes, ultrafast photoelectrochemical reactions triggered by pulsed laser excitation, etc. Such studies will benefit from the proven low-dose, buried-interface sensitivity, and diffraction-order selectivity established here, while leveraging the technique's compatibility with microsecond-scale acquisition. We view the current work as a foundational demonstration that paves the way for these future ultrafast applications.

Revision made: We added the following illustration into the revised SI as the **Supplementary Note 7** to make a clear clarification of the millisecond exposure time of PE-RSoXS in a single shot.

“In this work, the term “non-destructive” refers to the demonstrated ability of PE-RSoXS to acquire high-quality diffraction and spectroscopic data before any measurable X-ray radiation-induced chemical or structural changes occur in the sample. The 1 ms of single-shot exposure time used here was selected as an optimal trade-off between preserving the native state of the solid-liquid interfaces and ensuring sufficient signal-to-noise ratio for reliable data analysis, rather than as the minimum achievable exposure. PE-RSoXS was implemented with such a short single-shot exposure time (1 ms) to enable *operando* monitoring of electrochemical solid-

liquid interfaces while minimizing radiation dose ($< 0.01 \text{ mJ}\cdot\text{cm}^{-2}$ per frame) and preserving the native state of the nanopatterns. Although the technique and instrumentation are intrinsically capable of reaching as low as microsecond timescales, the present study focused on a well-characterized Ni LGNPs/OER model system with dynamics in the millisecond regime or slower ones to benchmark dimensional and chemical sensitivity at the buried interfaces under realistic electrocatalytic conditions. This methodological validation provides the foundation for future studies targeting ultrafast processes, such as microsecond electrochemical switching or laser-triggered photoelectrochemical reactions, where the full temporal potential of PE-RSoXS can be exploited.”

4. In line 254, the authors use simulations to reveal the potential-induced change in oxidation layer thickness of the Ni-based catalyst. However, there is a lack of direct experimental evidence to support this claim. Since the oxidation of Ni during the OER is generally irreversible, changes in the oxide layer thickness should be observable by electron microscopy. Therefore, additional experimental evidence is needed to support the PE-RSoXS observations.

Response: We thank the Reviewer for highlighting the importance of direct “visible” experimental evidence to support the potential-induced structural changes. To address this, we performed post-mortem atomic force microscopy (AFM) measurements on the Ni LGNPs after the electrochemical experiments, as shown in **Figure R1**.

Figure R1. Post-mortem AFM images of the Ni LGNPs on the SiN_x windows of the sample chip after electrocatalysis at **a**, OCP, **b**, 1.2 V and **c**, 1.6 V, showing the statistical line width and line edge roughness (LER) at each condition.

The AFM results show total width changes of the nanopatterns that closely follow the trend simulated by our finite-element method (FEM) based on PE-RSoXS results (OCP: 113.8 nm; 1.2 V: 113.4 nm; 1.6 V: 120.5 nm). However, in these AFM images, clear fluctuations are observed along the line edges. These fluctuations can be quantified using the concept of line edge roughness (LER), as described by Nealey *et al.* [*J. Appl. Cryst.* 49, 823–834 (2016)]. Meanwhile, *ex situ* microscopy techniques cannot directly measure the oxide layer thickness with sub-nanometer resolution. High-resolution TEM could, in principle, provide this information, but it requires to prepare cross-sections of the line gratings, which may potentially damage the sample. In contrast, PE-RSoXS enables *operando*, low-dose, and non-destructive measurements that quantitatively track oxide layer thickness and core-shell structural changes during active electrocatalysis. By exploiting resonant scattering and diffraction-order sensitivity, PE-RSoXS can distinguish shell and core contributions without relying on post-mortem assumptions, providing statistically representative results across over hundreds of nanopatterns under realistic electrochemical conditions, which highlights the significance to develop PE-RSoXS.

Revision made: We added **Figure R1** into the revised SI as the **Supplementary Fig. 8** to exhibit the Ni LGNPs after electrocatalysis and supplemented the following illustration into the revised manuscript to compare our simulation results and AFM results of the structural variations at each condition, highlighting the significance to develop PE-RSoXS.

“Besides, our simulated total width changes of the Ni LGNPs align with atomic force microscopy (AFM) results from the samples after being operated at the corresponding conditions (Supplementary Fig. 8), further clarifying the fidelity of our FEM-based simulations. However, in these AFM images, clear fluctuations are observed along the line edges. And *ex situ* microscopy techniques cannot directly measure the oxide layer thickness with sub-nanometer resolution. In contrast, by exploiting resonant scattering and diffraction-order sensitivity, PE-RSoXS can distinguish shell and core contributions without relying on post-mortem assumptions, providing statistically representative results across over hundreds of nanopatterns under realistic electrochemical conditions, which highlights the significance to develop PE-RSoXS.”

5. The authors provide a detailed discussion of PE-RSoXS principles and technical merits in the main text. While this information is valuable, moving it to the supplementary information would improve the manuscript’s focus and clarity. The main text should instead provide a concise and clear discussion that highlights the distinctive structural features revealed by PE-RSoXS and emphasizes the necessity of this technique for the study. This approach will attract greater interest from readers.

Response: We appreciate the Reviewer’s suggestion to improve the manuscript’s focus. We fully agree that a concise and clear discussion of the distinctive structural features revealed by PE-RSoXS, and the necessity of this technique will help engage a broader readership. We moved some PE-RSoXS principles and technical merits to the revised SI. Given that the primary novelty of this work lies in the development of a new *operando* X-ray methodology, we believe it is essential to retain concise core principles and working mechanism of PE-RSoXS in the main text. The “sample-as-optics” design, Bragg diffraction enhancement, and diffraction-order-dependent separation of bulk and interfacial signals are integral to the interpretation of the experimental results and to understanding why PE-RSoXS can achieve sub-nanometer structural sensitivity and millisecond temporal resolution under realistic electrochemical conditions. We hope to retain these contents to ensure that readers, particularly those outside the RSoXS community, can fully understand how the technique operates, what makes it unique, and why it enables capabilities beyond existing methods. Meanwhile, we have carefully revised the manuscript to accommodate the Reviewer’s suggestion for greater clarity and focus. In the Results and Discussion section, we have added a concise summary paragraph that explicitly highlights the distinctive structural features revealed by PE-RSoXS and emphasizes the necessity of this technique for the study.

Revision made: We have modified PE-RSoXS principles and the technical merits and moved some contents to the revise SI as **Supplementary Note 2**. We also added the following illustration to the end of the section “**Operando monitoring of electrochemical solid-liquid interfaces by PE-RSoXS**” in the revised manuscript.

“These results unveil the distinctive capability of PE-RSoXS to *operando* resolve subtle structural and chemical transformations at buried solid-liquid interfaces. The method quantitatively captures sub-nanometer dimensional changes in the Ni LGNPs during OER while simultaneously mapping oxidation-state gradients between the metallic Ni core and the evolving Ni(OH)₂/NiOOH shell through the FEM simulations on the diffraction-order-dependent scattering spectra. This dual sensitivity to geometry and chemistry, achieved under ultra-low X-ray doses with statistical averaging across hundreds of identical nanopattern units,

enables a direct correlation between redox transitions and sub-nanometer morphological reconstruction. Such interface-specific insights, particularly the separation of bulk and interfacial signals based on the core-shell structures, are essential for understanding electrocatalytic behaviors, which is still difficult to be obtained with conventional soft XAS or electron microscopy under realistic electrochemical conditions.”

6. The PE-RSoXS measurements require highly specialized sample preparation to produce well-ordered Ni line-grating nanopatterns. Although this approach is interesting, its broader applicability, particularly to conventional powder-form nanocatalysts, remains to be demonstrated.

Response: We understand the Reviewer’s concern regarding the specialized sample preparation for PE-RSoXS measurements. As we summarized in the beginning, RSoXS has been indeed successfully applied to powder electrocatalysts in recent high-impact studies [Yang et al. *J. Am. Chem. Soc.* 144, 8927–8931 (2022); Yang et al. *Nature* 614, 262–269 (2023); Yang et al. *Nat. Catal.* 8, 579–594 (2025)]. We emphasize that the current work uses nanopatterned electrodes as a deliberate model platform to establish and validate the PE-RSoXS methodology’s resolution limits, chemical sensitivity, and ability to decouple bulk and interfacial signals based on the working principles of RSoXS, which are difficult to benchmark quantitatively in disordered powder systems. The specialized nanopatterned electrodes in this study are not a limitation, but a methodological enabler, providing the controlled geometry and optical modulation necessary to validate PE-RSoXS. As the Reviewer mentioned, powders remain the dominant form for practical electrocatalysis. We expect that mechanistic insights derived from the model systems are directly translatable to conventional powder electrocatalysts, thereby underscoring the wider applicability of the technique; this perspective will be incorporated into the Conclusion section.

Revision made: We have added the following statement in the “**Conclusion**” of the revised manuscript to emphasize the generality of PE-RSoXS and outline future applications to restructuring catalysts beyond Ni-based catalysts.

“The PE-RSoXS methodology is broadly applicable beyond the Ni-based system studied in this work, providing *operando* access to coupled structural and chemical dynamics in a wide range of catalytic materials that undergo significant restructuring during operation. By tuning to the relevant absorption edges of the focused elements, PE-RSoXS can selectively enhance scattering signals from specific chemical states, enabling real-time correlation between sub-nanometer structural changes and active-phase evolution. Its compatibility with identical architectures including powder- and pattern-based systems, low-dose operation, and high sensitivity to the buried electrochemical interfaces make it a versatile tool for probing diverse catalysts under realistic electrochemical conditions, with strong potential to address key questions in catalysis, energy conversion, and materials science.”

Thanks again for the valuable comments and suggestions from the Reviewer. We have carefully addressed each point to meet the high standards expected.

Reviewer: 2

The manuscript describes an implementation of the emerging RSoXS scattering method that exploits a Bragg structure sample to gain strong sensitivity to the composition within it. The new implementation is coined “PE-RSoXS.” It is demonstrated in a Nickel electrochemical oxidation experiment, and different optical constants are pulled out that describe different shell layers obtained at different electrochemical potentials. The analysis framework features a full Maxwell simulation beyond typical Born approximation approaches.

The manuscript reflects important advances and should be considered for publication in Nature Communications, provided the issues identified in review are addressed. Although this reviewer will provide extensive feedback for the author team (see below), the volume of feedback does not reflect manuscript quality. There are issues in presentation and context, and some technical issues that I will highlight in detail. Provided the editors agree, I look forward to continued consideration of this article for publication in this fine journal.

Response: We sincerely thank the Reviewer for the positive and encouraging assessment of our work and for recognizing the advances presented in the development and implementation of the PE-RSoXS methodology. We are pleased that the novelty of exploiting a Bragg-structured sample for enhanced compositional sensitivity, the *operando* demonstration in Ni electrochemical oxidation, and the application of a full Maxwell simulation framework beyond the typical Born approximation were well received. We also greatly appreciate the Reviewer’s willingness to provide extensive feedback and understand that the volume of comments reflects a commitment to improving the clarity, context, and technical rigor of the manuscript. We have carefully considered all points raised by the Reviewer and have made substantial point-by-point revisions to address issues in presentation, contextualization, and technical details below. We believe that these changes have improved the manuscript’s readability, completeness, and impact, and we look forward to the continued consideration of our article for publication in *Nature Communications*.

- Overarchingly, the centerpiece of the manuscript describes a technique that is usually called “diffraction anomalous fine structure,” or DAFS. This technique is actually very old. It has been described, in some form, since 1956 - Y. Cauchois (1956) — “Distribution spectrale observée dans une région d’absorption propre de divers cristaux”, C. R. Acad. Sci. Paris 242, 100–102. The idea was extended in work by other authors in the mid-70s, and re-emerged as a method named DAFS in the surface science era of the early-to-mid-90s, with a flagship publication led by Joe Woicik who is now at NIST: Stragier, H., Cross, J. O., Rehr, J. J., Sorensen, L. B., Bouldin, C. E., & Woicik, J. C. (1992). Diffraction anomalous fine structure: A new x-ray structural technique. *Phys. Rev. Lett.*, 69, 3064–3067. doi:10.1103/PhysRevLett.69.3064.

None of this rich history is acknowledged in the manuscript under review, nor are there any explanations of how the current manuscript goes beyond. In this reviewer’s eyes, there are at least two claims of novelty / advance for the manuscript under review: 1) there are very few implementation of DAFS with soft X-rays on larger Bragg structures, and 2) the simulations in the manuscript under review and the method of obtaining optical constants are extremely sophisticated compared to most previous DAFS examples, and particularly well-suited to soft X-rays given their high likelihood of multiple scattering. The authors should acknowledge the rich history of DAFS (a nice 2018 review is here <https://arxiv.org/pdf/cond-mat/0309624> but I don’t know if it was published outside arxiv), and then explain their advances.

Response: We thank the Reviewer for pointing out the historical context and foundational relevance of Diffraction Anomalous Fine Structure (DAFS) to our work. We agree that DAFS has a rich history, and we appreciate the opportunity to clarify how our *operando* PE-RSoXS approach relates to, extends, and differentiates itself from traditional DAFS.

DAFS is a synchrotron-based X-ray technique that combines Bragg diffraction with energy-dependent anomalous scattering near an absorption edge to extract local structural and chemical information about specific species within a periodic structure. In essence, DAFS merges the element-specificity of X-ray absorption spectroscopy (XAS) with the spatial selectivity of diffraction, enabling the probing of atomic environments at crystallographically distinct sites.

Historically, this concept was first noted by Y. Cauchois [*C. R. Acad. Sci.* 242, 100–102 (1956)], who observed spectral variations in diffraction intensities near absorption edges. The method was extended in the 1970s and formalized in the early 1990s, notably in the seminal work by Stragier et al. [*Phys. Rev. Lett.* 69, 3064–3067 (1992)], which coined the term "DAFS" and demonstrated its potential for probing local structure via diffraction intensity modulations. Since then, DAFS has been applied primarily in hard X-ray regimes to study bulk crystals, epitaxial films, and surface reconstructions, with relatively few implementations in the soft X-ray range due to challenges such as strong absorption, multiple scattering, and limited penetration depth.

Our PE-RSoXS shares conceptual similarities with DAFS in that both techniques exploit energy-dependent diffraction intensity variations near an absorption edge to obtain chemical and structural information. Specifically, like DAFS, PE-RSoXS derives spectroscopic profiles from diffraction intensities across a photon energy range, allowing element-specific probing of local environments. Both approaches can separate contributions from distinct structural motifs based on diffraction order or reciprocal space position. However, our work introduces several key conceptual and methodological advances that go beyond traditional DAFS:

(1) “Sample-as-Optics” Architecture with Engineered Nanopatterns

Traditional DAFS relies on naturally occurring or epitaxially grown periodic structures (e.g., atomic planes in a crystal). In contrast, PE-RSoXS deliberately incorporates nanoengineered line-grating nanopatterns into the sample design, which act as diffractive optical elements in the Fresnel (near-field) regime. This allows coherent amplification of weak resonant scattering signals from buried interfaces, achieving N^2 scaling in diffraction intensity and enabling *operando* measurements with sub-nanometer dimensional sensitivity.

(2) Soft X-ray Implementation for Buried Solid-Liquid Interfaces

While there have been few soft X-ray DAFS implementations, they typically target large Bragg structures in vacuum or surface science contexts. PE-RSoXS operates at soft X-ray energies (e.g., Ni L-edge at around 852 eV) under *operando* electrochemical conditions in a liquid flow cell, overcoming challenges of strong absorption and multiple scattering by leveraging the length-scale match between soft X-ray wavelengths, nanopattern periodicity, and interfacial structures. This enables direct, non-destructive probing of buried solid-liquid interfaces during active catalysis, which is a regime largely inaccessible to traditional DAFS.

(3) Diffraction-Order-Dependent Chemical Sensitivity

In PE-RSoXS, diffraction order dependency is engineered via the duty cycle and symmetry of the nanopatterns to selectively suppress or enhance certain orders. This allows us to decouple interfacial signals from bulk contributions using 50% duty cycle patterned structures, where even-order diffraction intensities are highly sensitive to subtle interfacial inhomogeneities, while odd-order signals reflect bulk properties. This level of order-selective chemical sensitivity is not generally exploited in conventional DAFS.

(4) Advanced FEM-Based Simulations Coupled to Bayesian Optimization

Our work integrates finite-element Maxwell solvers to rigorously model multiple scattering and near-field effects in sub-wavelength structures, which is a necessity for soft X-ray regimes where multiple scattering is significant. We extract energy-dependent optical constants for both

core and shell regions, enabling quantitative reconstruction of oxidation-state gradients and structural changes. This simulation–experiment synergy is far more sophisticated than most previous DAFS implementations, which often rely on simplified kinematic diffraction models.

Therefore, while PE-RSoXS builds upon the foundational principle of DAFS, it still represents a new class of interface-specific metrology.

Revision made: We have acknowledged DAFS in the “**Working principle of PE-RSoXS**” of the revised manuscript, citing the foundational works [Cauchois et al. *C. R. Acad. Sci. Paris* 242, 100–102 (1956); Stragier et al. *Phys. Rev. Lett.* 69, 3064–3067 (1992); Renevier et al. *Synchrotron Radiat.* 10, 435–444 (2003)] and explicitly positioning PE-RSoXS as a conceptual and technical evolution of DAFS tailored for *operando* soft X-ray studies of buried electrochemical interfaces in the **Supplementary Note 2** (see below) of the revised SI.

“While diffraction anomalous fine structure (DAFS) has been widely applied in hard X-ray regimes for bulk crystals and epitaxial films¹⁹, its implementation in soft X-rays has been limited due to strong absorption and multiple scattering effects. Compared to traditional DAFS, PE-RSoXS extends this principle by integrating engineered periodic nanopatterns directly into the sample design. Well-ordered line-grating nanopatterns with uniform rectangular geometry were constructed to enhance scattering sensitivity. When the incident soft X-rays are aligned perpendicularly to the grating lines, the periodic structures produce a series of diffraction patterns, as known as “Bragg spots”, recorded on a charge-coupled device (CCD) detector.

A unique advantage of PE-RSoXS is the natural alignment of length scales among the soft X-ray wavelength, the engineered nanopatterns, and the interfacial phenomena of interests^{20,21}. As a proof of concept, at the Ni L-edge (around 852 eV), the soft X-ray wavelength is around 1.5 nm, which is comparable to the length scale of buried interfacial structures as usually shown in reconstructions and chemical gradients at the solid-liquid boundaries during electrocatalysis^{22–24}. Meanwhile, the precisely engineered nanopatterns with the width size in order of tens of nanometers, provide a periodic optical modulation that coherently amplifies weak scattering intensity changes of the line-gratings (form factors), especially from the buried electrochemical interfaces^{25,26}. This length-scale alignment is a distinctive advantage in optical and infrared methods, which conventionally lack sufficient dimensional sensitivity^{27,28}. Besides, hard X-rays have much shorter wavelengths compared to soft X-rays, and they would largely reduce the energy resonance and dimensionally modulated contrast^{29,30}. PE-RSoXS bridges this gap and enables *operando* access to structural and chemical dynamics with nanometer to sub-nanometer resolution and element-specific sensitivity at solid-liquid interfaces, respectively.”

- The reviewers make multiple claims of interface sensitivity - the word is mentioned 45 times in the article! But there is nothing especially interface sensitive about RSoXS or PE-RSoXS. The RSoXS technique is FORM FACTOR sensitive. When the form factor includes an interface, we can of course be sensitive to it, and different placements of the interface within the form factor (and within the structure factor it is convoluted with) will result in different levels of sensitivity.

Yes, in the implementation here, the authors demonstrate sensitivity to a shell in a core-shell structure (this is the right language, not “interface sensitive”), but there will also be sensitivity

to other parts of the structure that are not interfacial. The technique will be excruciatingly sensitive to the density and composition of the solid core, for example.

The reason this reviewer is so adamant about the language used here is that the “interface sensitive” claims are misleading and they echo misleading claims made about RSoXS about 10-15 years ago that have (in this reviewer’s opinion) sown confusion and stunted the growth of the technique. Much early work on the RSoXS of organic electronics and organic photovoltaics (OPVs) in particular made claims about special sensitivity to interfacial composition and orientation. The language used then (and now, in the article under review!) was extremely similar to the language of nonlinear optics techniques such as sum frequency generation (SFG) (which was emerging and highly visible at the time), in which symmetry breaking at interfaces truly affects spectroscopic selection rules, allowing spectra to be observed that isolate signals from the interface. That is NOT what is happening in RSoXS – scattering sees everything, not just interfaces, and RSoXS is frequently bulk-dominated. But this article strongly implies some kind of special sensitivity to interfaces with the 45 mentions of the word “interface” and in fact these authors use “symmetry breaking” language to explain their results in lines 127-128, when what is meant is that subtle perturbations in the form factor can have strong effects on order expression. But these perturbations don’t necessarily have to be at interfaces, and RSoXS is not sensitive to symmetry breaking specifically, just the form factor. The authors should remove their misleading language about interface sensitivity and symmetry breaking. If they would like to include it, they should include the context that the technique is sensitive to interfaces *when you know there is an interface, and you know where it is, and you can model it*. That is what is required. It’s also the same way the RSoXS is sensitive to aspects of structure other than interfaces.

Response: We appreciate the Reviewer’s thoughtful and technically grounded critique. We fully agree that RSoXS and PE-RSoXS are fundamentally form factor sensitive techniques. Any sensitivity to interfaces arises only when an interface is explicitly modeled as a distinct structural region within the sample, and its boundaries are incorporated into the form factor. In our study, the interfacial region is defined as the shell in a core-shell geometry. This shell lies between the metallic Ni core and the surrounding electrolyte, and it is within this shell that electrochemical reactions take place. Structural and chemical variations in this shell region affect the form factor, which in turn modulates the scattering signal. We clarified this definition in the revised manuscript.

We also understand the Reviewer’s concern regarding the use of “interface sensitivity”, which has been historically misapplied in the RSoXS literature. In response, we have carefully reviewed all instances of this language and revised them to more precisely describe the mechanism of contrast generation as form factor sensitivity, particularly in relation to well-defined structural subregions such as shells. Additionally, we have removed or rephrased uses of the term “symmetry breaking” to avoid unintended associations with nonlinear optical selection rules. In our revised text, we now describe these effects as deviations from structural symmetry in the form factor, which modulate the scattering intensity of diffraction peaks.

We believe these revisions clarify the physical interpretation of PE-RSoXS and align the terminology with established principles in the X-ray scattering community. We are grateful for the Reviewer’s historical insight and constructive suggestions, which have improved the accuracy and rigor of our manuscript.

Revision made: Throughout the revised manuscript, we have replaced broad “interface sensitivity” phrasing with the concept of “form-factor sensitivity to interfacial regions” or analogues, making clear that this sensitivity is engineered via sample design and confirmed through modeling. We have also added the following statements to the revised manuscript and SI respectively to clarify that PE-RSoXS is not inherently interface-specific, but our design and

measurement strategy make it exceptionally sensitive to buried electrochemical solid-liquid interfaces in this study.

In the “**Abstract**”: “...PE-RSoXS opens new pathways to **operando** exploration of chemical evolution and sub-nanometer dimensional variations in nanostructured electrochemical systems. This method is non-destructive, highly efficient, and element-specific, allowing for the detection of chemical and dimensional dynamics at reaction fronts when appropriately modeled, due to its sensitivity to form factors.”

In the “**Working principle of PE-RSoXS**” section: “In our implementation, the interfacial region is defined as the shell layer in a Ni core-shell structure. This region evolves chemically and structurally during electrochemical OER, and its presence modifies the form factor, which is the basis for scattering contrast in PE-RSoXS.”

In the “**Operando monitoring of electrochemical solid-liquid interfaces by PE-RSoXS**” section: “The evolution of the shell (interfacial) region is modeled as a change in form factor due to the oxidation-state-dependent optical constants. The observed spectral changes are not exclusive to interfaces but rather reflect the local chemical environment within the shell region defined in the form factor.”

Supplementary Note 3 in the revised SI “We note that RSoXS, and by extension to PE-RSoXS, is fundamentally form-factor sensitive. The scattering intensity arises from the electron density distribution within the scattering object, convoluted with its structural periodicity. The PE-RSoXS technique is not inherently interface-specific, nor does it operate under spectroscopic selection rules like those in nonlinear optics. In our implementation, we deliberately designed Ni line-grating nanopatterns (Ni LGNPs) for PE-RSoXS measurements and they formed a core-shell geometry so that the buried electrochemical solid-liquid interfaces contribute strongly to the form-factor contrast. Perturbations in this shell region under operating conditions, which were confirmed through FEM-simulations, selectively influence certain diffraction orders, enabling us to isolate and *operando* monitor interfacial changes. This approach could be equally applied to emphasize bulk or non-interfacial regions, depending on the sample design and modeling strategy.”

- The above concern brings me to an adjacent point, which is that the authors have not demonstrated the “spatial resolution” that they mention 19 times in the manuscript. What they have demonstrated is fine sensitivity to shell thickness in a core-shell structure. Which, in this reviewer’s opinion, should certainly be enough for publication in Nat Comm if the authors are willing to soften their claims. But “spatial resolution” implies that they can identify **where** that material is within the structure. Instead, the authors have assumed that it is on the sidewall of the solid structure. That is, of course, a perfectly valid assumption given the electrochemical nature of the experiment, but it does not demonstrate spatial resolution.

To demonstrate spatial resolution, the authors could consider simulating what happens if the shell were “free floating” within the electrolyte-filled part of the Bragg structure, spaced by various distances from the wall. As unphysical as those models would be, that could potentially demonstrate spatial resolution. I can assert from personal experience that scattering from Bragg structures is typically far less sensitive to specific placement within the Bragg structure (“spatial resolution”) than it is sensitive to the specific volume fractions (shell thickness) and their indices (shell and core and matrix composition), but I would defer to simulation results from the authors if they showed otherwise.

Response: We appreciate the Reviewer’s thoughtful feedback on the distinction between spatial resolution and sensitivity to shell thickness, and we understand the concern regarding the strength of our claims. We followed the Reviewer’s suggestion and performed simulations

to examine how scattering profiles change when the shell is artificially displaced within the electrolyte-filled groove of the nanopattern, including unphysical but informative scenarios where the shell is not in direct contact with the core.

To investigate the influence of material placement, we employed a simplified model of a Ni line-grating with periodic boundary conditions applied laterally, as shown in **Figure R2**. The simulation was performed using an incident photon energy of 852 eV. The grating cross-section is rectangular, and two smaller rectangular features matching the Ni grating in height, were added to represent the oxide layer. The geometric dimensions of all components were kept constant throughout the study. In configurations A and B, the oxide layers are symmetrically positioned at distances of 15 nm and 5 nm, respectively, from the Ni grating bulk. In configurations C and D, the oxide layers retain the same placement as configuration A, while the Ni grating is shifted laterally by 5 nm to the right and left, respectively. For all configurations, the resulting diffraction peak intensities are shown in **Figure R2b**. The pronounced differences in intensity across configurations demonstrate that the method is highly sensitive to the relative placement of the constituent materials.

Importantly, PE-RSoXS operates on principles analogous to protein crystallography, where diffraction from periodic units enables reconstruction of real-space features with sub-nanometer to nanometer-scale precision. In our case, the periodic nanopatterns define a structural unit cell, and the spatial distribution of chemical species within that unit cell modulates the scattering form factor in a position-sensitive manner. This coherence-based sensitivity to the distribution and geometry of resonant materials enables spatial resolution at sub-nanometer length scales, as supported by both the experimental data and the FEM modeling.

Figure R2. a, Four configurations (A to D) of a Ni metal core with oxidized Ni shells at different placements for diffraction intensity simulation. **b**, The simulated diffraction peak intensities of the configurations A to D at 852 eV.

Revision made: We have revised the manuscript to clarify this conceptual framework and have softened language around “spatial resolution” where appropriate. The updated text also emphasizes that the demonstrated resolution is possible because the location of the shell is well-defined and constrained by the model geometry, and that such spatial information is resolvable through order-dependent scattering when supported by rigorous simulations. We added **Figure R2** into the revised SI as the **Supplementary Fig. 13**. In the revised manuscript, we added the following illustrations in the section “**Theoretical FEM-based Simulations of PE-RSoXS**” of the revised manuscript.

“To address the concept of spatial resolution more rigorously, we conducted additional simulations (Supplementary Fig. 13). The oxidized Ni shell region was displaced with distances from Ni metal core without physical contact. These simulations reveal that the diffraction signal is not only sensitive to shell thickness and composition, but also to the position of the shell within the grating period, confirming that PE-RSoXS can resolve spatial placement within the structural unit cell. By simulating unphysical scenarios in which the shell is “floating” at varying positions within the groove, we demonstrate that the diffraction intensities change depending on shell placement, indicating the system’s sensitivity to spatial localization of chemical features.

As a result, our simulations demonstrate the characteristic features of PE-RSoXS in unveiling interfaces: (i) enhanced scattering signal from buried interfaces by statistically identical nanopatterns (**form factors**) and decoupled the interfacial signals from the bulk material responses; (ii) high scattering sensitivity to sub-nanometer structural variations; (iii) site-specific sensitivity of elemental oxidation state by order dependency; (iv) **achievable spatial resolution within the unit cell when the geometry is well-defined.**”

- There is a significant missed opportunity to structure the narrative here with deeper fundamentals in the presentation of the “scattering derived spectra.” These appear to be [I integrated across a peak] vs. energy, presented in Fig 2 d-e-f. On lines 190-191 these spectra are directly compared to X-ray absorption spectra data, and described as “comparable.” But it is incorrect to compare them because [I integrated across a peak] is the result of *differences between indices* of the volumes within the scattering structure, and it includes the delta parts of those indices. So, there is dispersion baked into that “spectra”, and even if you could isolate beta, it would be a difference spectrum between nickel beta and nickel oxide beta (electrolyte beta would be reasonably flat, but sloped). The authors should remove this comparison to XAS, and more properly acknowledge the contributions to I vs. energy.

But there is an even stronger way to do that, and that is to lean into the concept of the total scattering invariant (TSI). The TSI describes the integrated scattering across all q and in all 3 dimensions; for RSoXS patterns an approximation frequently called “integrated scattering intensity” (ISI) is used by integrating $I \cdot q^2$ (sometimes q-bounded or feature-bounded, and sometimes for all measurable q) vs energy. The authors could consider presenting Fig 2 d-e-f as feature-bounded ISI vs. energy. The reason this would be valuable is that, in principle, TSI / ISI should be directly proportional to the sum of the combinatorial pairwise binary contrasts $(\Delta(\text{beta})^{**2} + \Delta(\text{delta})^{**2})$ weighted by their volume fraction products (see Collins and Gann 2021 Eq 8 and Eq 15; it can be extended to any number of components). Quantitative agreement of the energy-dependent ISI vs. energy and expectation contrast was robustly demonstrated in Ferron, Pope, and Collins PRL 2017, and it underpins the physics of RSoXS and of the manuscript under review. The Maxwell solver will reproduce this physics but the authors should more robustly acknowledge it, as they deal with I vs. energy much earlier than they introduce n, delta, beta, etc, when they arise from a common framework.

Response: We appreciate the Reviewer’s insightful and technically rigorous comment. We agree that scattering-derived intensity profiles (Fig. 2d–f) are fundamentally different from

conventional X-ray absorption spectra (XAS), and we acknowledge that describing them as “comparable” may be misleading without proper qualification. We emphasize that the scattering-derived spectra represent energy-dependent diffraction intensities that emerge from spatially modulated optical contrast within the periodic nanostructure. These spectra reflect the combined influence of the dispersion (δ) and absorption (β) components of the complex refractive index ($n = 1 - \delta + i\beta$) and cannot be interpreted as direct analogs of XAS. We have removed the word “comparable” and clarified the distinction in the revised manuscript.

We also recognize the reviewer’s suggestion to frame the analysis more fundamentally in terms of the total scattering invariant (TSI) or feature-bounded integrated scattering intensity (ISI). While ISI-based analysis is highly effective in binary systems or where phase separation is well defined, our system involves a graded and non-binary core-shell geometry with subtle oxidation-state-dependent contrasts. In such a configuration, ISI collapses the energy dependence of scattering into a single scalar value per energy and obscures the spatial distribution of contrast. Therefore, we think ISI is not ideally suited for our *operando* analysis. Instead, our approach leverages a forward-modeling framework based on FEM, where the energy-dependent diffraction intensities across multiple diffraction orders are simultaneously fit to extract the underlying optical constants (δ and β) and geometry of the core-shell structures. Notably, the β component extracted via simulation (see Fig. 3d) is directly related to the absorption component of the optical response and does qualitatively reproduce Ni L-edge features, consistent with XAS reference data. This connection is established not through a scalar ISI, but via spatially resolved modeling that reproduces the order-dependent scattering profiles as a function of energy.

Revision made: We have changed the Y-axis title of all scattering profiles to “Diffraction Intensity” in the revised manuscript and SI and removed the previous comparison to XAS. We also have clarified that these are scattering-derived signatures and added the following demonstrations into the revised manuscript. We also acknowledged the foundational theoretical framework that underpins this physics with new references in the revised manuscript.

Page 5 in the section “***Operando* monitoring of electrochemical solid-liquid interfaces by PE-RSoXS**” of the revised manuscript:

“...producing energy-dependent scattering profiles. While these spectra resemble XAS in that they exhibit Ni L-edge features, they are not directly equivalent. The measured diffraction intensities result from spatially modulated differences in the complex refractive index ($n = 1 - \delta + i\beta$) and reflect both absorption (β) and dispersion (δ) contributions.”

Page 6 in the section “***Operando* monitoring of electrochemical solid-liquid interfaces by PE-RSoXS**” of the revised manuscript:

“For $\beta(E)$, which corresponds to the absorptive component of the optical response, the simulation captures key features of Ni L-edge absorption in the shell region. These features are not direct XAS measurements but rather arise from the fitted optical constants that best reproduce the energy-dependent scattering intensity across all diffraction orders.”

Supplementary Note 6 in the revised SI:

“We also considered the possibility of presenting integrated scattering intensity (ISI) curves to reflect the total contrast within the system. However, because our system features a graded and non-binary core-shell geometry with subtle variations in oxidation state, ISI-based methods (which typically integrate $I \cdot q^2$ across all or bounded q) are not ideally suited for resolving localized compositional changes. Instead, we adopted a FEM-modeling approach, in which the energy-dependent diffraction intensities across multiple orders are simultaneously fit to extract the geometry and optical constants of the shell. This approach enables location-specific reconstruction of $\beta(E)$ (the absorption component), which closely reflects the X-ray absorption

behavior of the oxidized Ni species. This framework is consistent with the physical basis reported previously^{4,32}.”

The authors’ mentions of “numbers of scatterers N ” on lines 155-156 and the sensitivity to N squared are a confusing and less general recapitulation of the physics I mention above. They are misleading in stating that “conventional scattering” has linear sensitivity to the number of scatterers. All SAS has $\Delta(\rho)^2$ magnitude; the number of scattering objects increases the scattering intensity linearly in the dilute form factor limit, whereas in a Bragg structure you have a *really* strong reporter of $\Delta(\rho)^2$. Perhaps that’s what the authors intended to say but the concepts are muddled here.

Response: We thank the Reviewer for pointing out that our current discussion of “number of scatterers N ” and N^2 sensitivity is insufficiently general and could be misleading in its comparison to “conventional scattering.” We recognize that the comment may stem from a conflation of two different physical concepts:

(1) the well-known contrast dependence of scattering intensity via $\Delta\rho^2$ or Δn^2 , which governs the magnitude of the scattering signal, and (2) the coherent enhancement due to periodicity, which leads to an intensity scaling of N^2 , and the number of periodic units contributes constructively to phase.

Our reference to “ N squared” sensitivity was not meant to describe the typical contrast dependence in small-angle scattering, but rather to highlight a fundamental principle of pattern enhancement (Bragg Diffraction) in coherent scattering systems. Specifically, in PE-RSoXS, the nanopatterned structure acts as a diffraction grating where each identical unit contributes a scattering amplitude A , and the total amplitude scales as $N \cdot A$, when the scattered waves are coherently added in phase. As a result, the intensity scales as $I \propto [N \cdot A]^2 = N^2 \cdot |A|^2$

Revision made: We have added the following demonstration into the revised SI as the **Supplementary Note 5**.

“In our patterned system, the coherent scattering amplitude from each identical unit adds in phase, resulting in a total amplitude that scales as $N \cdot A$, where N is the number of the identical units. Consequently, the scattered intensity scales as N^2 , in contrast to the linear scaling typically observed in systems with randomly distributed or dilute scatterers. This coherent enhancement is distinct from the contrast-driven scattering magnitude ($\Delta\rho^2$ or Δn^2) discussed in small-angle scattering and is central to the concept of PE-RSoXS introduced in this work. In our experiment, with approximately 400 identical Ni line-grating units, it leads to a five-order-of-magnitude enhancement in scattering signal, enabling ultrahigh sensitivity to interfacial changes.”

- The authors should moderate their language and results that suggest spatial resolution beyond the diffraction limit, and they need to be clear about their interaction with this limit. Spatial resolution beyond the diffraction limit is implied by their language: “atomic scale resolution” is used line 73. But the wavelength here would be ~ 1.5 nm, with a diffraction limit of 0.75 nm. The PE-RSoXS technique – like every other SAS technique that exists - should not be capable of real spatial resolution beyond that diffraction limit. But, the authors do demonstrate major differences in the I vs. energy simulation for 0.2 nm increments in shell thickness. This does NOT demonstrate 0.2 nm spatial resolution, it goes back to my earlier point that we are exquisitely sensitive to volume fraction of the shell (or an alternative way of thinking is the effective medium index within diffraction limit volumes), and less sensitive to where the shell is placed within the structure (what we would call “spatial resolution”). If we assume or have robust proof that the shell must be attached to the sidewall, then we have an \sim ersatz sub-

diffraction-limit spatial resolution, but the assumption about the location is doing all the heavy lifting in that result. I strongly recommend the authors reconsider their language and directly address their interaction with the diffraction limit lest they mislead readers.

Response: We thank the Reviewer for raising this important point. We fully agree that PE-RSoXS, like all wave-based scattering techniques, is subject to the diffraction limit, which in our case is defined by the ~1.5 nm wavelength of the soft X-rays at the Ni L-edge, corresponding to a theoretical resolution limit of ~0.75 nm. Our previous use of the phrase “atomic-scale resolution” might have implied sub-diffraction imaging capability, which was not our intention.

We have revised this language throughout the manuscript to refer instead to “sub-nanometer structural sensitivity”, which more accurately reflects the nature of the technique. Specifically, the sensitivity to small variations in shell thickness (e.g., 0.2 nm increments from our FEM simulations) arises not from direct spatial resolution, but from the impact these variations have on the energy-dependent scattering form factor, as coherently probed across multiple diffraction orders. These changes are detectable within the diffraction-limited spatial scale, and they manifest through their influence on the diffraction intensity, not through direct imaging of features below half of the wavelength of the incident X-ray.

We also appreciate the opportunity to clarify that this approach is fundamentally model-based, in line with the principles of protein crystallography, where structural features are inferred by analyzing diffraction from periodic units. In both techniques, resolution is ultimately limited by the wavelength, but precise information about sub-nanometer features can still be extracted through coherent interference and accurate modeling of the internal structures within the unit cell. We have now made this analogy explicit in the revised manuscript and clarified our interaction with the diffraction limit.

Revision made: In the revised manuscript and SI, we have changed the terminology of “atomic scales” to “sub-nanometer” and made the following revisions.

In the “Introduction” section: “...detection of electrochemical interfaces under *operando* conditions at the sub-nanometer scales in the soft X-ray diffraction limit.”

In the “*Operando* monitoring of electrochemical solid-liquid interfaces by PE-RSoXS” section of the revised manuscript: “The spatial resolution of PE-RSoXS is fundamentally limited by the X-ray wavelength (~1.5 nm), resulting in a diffraction-limited resolution of approximately 0.75 nm. Our ability to detect smaller structural changes (e.g., 0.2 nm in shell thickness) arises not from direct imaging beyond this limit, but from the sensitivity of diffraction intensities to such variations within the periodic structure.”

- Aside from my points above, the introduction is otherwise very accessible and explains some key jargon pieces such as the meaning of “near field.” However, the description of the modeling – a centerpiece of the manuscript – is extremely confusing. Some questions that came up during my reading:

- . what is the “energy offset”? I’m guessing it’s to accommodate differences between the energy of the beamline at RSoXS measurement and the energies of the NEXAFS-derived library index used for the core. But this is never explicitly stated in the body or methods.
- . was the shell thickness floated in the fit? This is never explicitly stated.
- . I’m quite surprised at how nicely smooth the delta and beta DAFS extractions are. Did the authors use tricks to keep them smooth – like a thickness constraint or the use of the previous energy’s index as an initial guess for the next? These kinds of fittings are prone to local minima.

Response: We thank the Reviewer for these detailed questions regarding our modeling approach, which is indeed a centerpiece of the manuscript. We agree that the current description in the Methods section is insufficiently explicit on some points, and we have revised the text to make our fitting procedure and assumptions fully clear.

1. Energy Offset

The “energy offset” refers to a small calibration difference between the beamline energy scale used in PE-RSoXS experiments and the NEXAFS-derived reference data used to calculate the optical constants (δ and β) for the core materials. This offset accounts for slight misalignments in absolute photon energy (~ 0.2 – 0.3 eV) due to instrument calibration or beamline drift. The offset is applied during the fitting of the core’s optical constants to align features in $\beta(E)$ with those observed experimentally.

2. Shell Thickness Floating in the Fit

Yes, the shell thickness was treated as a floating parameter in the Bayesian optimization used for the FEM-based simulation of the experimental scattering-derived spectra. It was independently optimized for each applied potential (OCP, 1.2 V, and 1.6 V), alongside other parameters such as the shell’s optical constants and the roughness factor.

3. How are $\delta(E)$ and $\beta(E)$ so smooth? Were constraints used?

We agree that fitting energy-dependent optical constants over many energies can be prone to local minima. To ensure stable and physically meaningful results, we used a multi-Gaussian parameterization of $\beta(E)$, and derived $\delta(E)$ via Kramers–Kronig (KK) transformation. Additionally, the fit was performed sequentially across photon energies using the previous energy’s result as the initial guess to improve convergence and smoothness. This approach avoids unphysical oscillations and helps guide the optimizer through a continuous solution space.

Revision made: In the revised manuscript, we have added the following descriptions to the “**Methods**” section of the revised manuscript.

Energy offset: “The energy offset accounts for small calibration differences between the incident photon energy at the RSoXS beamline and the reference NEXAFS data used to compute the optical constants. This typically ranged between -0.2 and -0.3 eV and was used to align characteristic absorption features of the core.”

Shell thickness floating in the fitting: “The shell thickness was treated as a floating parameter and optimized independently for each potential condition using Bayesian optimization. It was varied along with the shell optical constants, Debye–Waller roughness, and core energy offset.”

“To improve smoothness and convergence, the fitting process was seeded using the optimized solution at the previous energy step as the initial guess for the next energy point. The optical constants $\beta(E)$ were parameterized using a sum of Gaussian functions, and $\delta(E)$ was calculated via Kramers–Kronig relations, ensuring that both components followed physically meaningful trends. This regularization mitigated local minima and produced smooth, interpretable spectral features.”

- The authors should acknowledge some limitations around model uniqueness. Importantly, the simulations are done on structures that perfect, and the real structures are not. An imperfection that strongly affects order expression (the principal thing that is fit here) is interfacial width / roughness, which is strongly convoluted with sidewall angle in q_{xy} scattering. The authors assume that the electrochemically converted material and its two interfaces are sub-atomically smooth. Any line-edge roughness or shell thickness inhomogeneity (and there is usually a significant amount with solid-state electrochemistry) in the core or shell will diminish higher

peak orders relative to lower ones and potentially could be misfit. I'm not suggesting the authors need more parameters for their model, but they should acknowledge this limitation and discuss its potential effects on the results.

Response: We thank the Reviewer for this important observation. We fully agree that real nanostructures inevitably contain imperfections, including sidewall angle variability, line-edge roughness, shell thickness inhomogeneity, etc., which are not captured in our idealized FEM models. Our modeling framework is intentionally simplified to balance interpretability, computational feasibility, and the risk of overfitting. However, we recognize that model uniqueness is inherently limited by such structural assumptions, and that interfacial roughness or inhomogeneity could affect the expression of higher diffraction orders, potentially leading to biased extraction of shell thickness or optical constants

We included a Debye–Waller-type roughness factor to account for some degree of order suppression, but we acknowledge that this is a simplified correction that cannot fully represent complex morphological variability. We have now explicitly stated this limitation in the Methods section and added a discussion of its possible effects in the Discussion portion of the revised manuscript. We have also clarified that while our model provides a consistent and physically interpretable fit to the experimental data across multiple diffraction orders, the absolute values of extracted parameters should be interpreted within the context of these modeling assumptions.

Revision made: In the “*Operando monitoring of electrochemical solid-liquid interfaces by PE-RSoXS*” section of the revised manuscript, we added “**While our FEM-based simulations provide strong agreement with experimental data across multiple diffraction orders, the model assumes idealized core-shell structures with sharp, uniform interfaces. Interfacial roughness, line-edge disorder, and thickness inhomogeneity, especially common in solid-state electrochemical systems, may reduce the intensity of higher-order diffraction peaks and influence parameter extraction. These effects are partially accounted for using a Debye–Waller factor (see Methods), but residual mismatch or misfit could still arise from structural imperfections not explicitly modeled (Supplementary Fig. 7 and Supplementary Note 8).**”

In the section of “**Methods**”, we added “**Roughness was incorporated using a Debye–Waller factor correction to model the attenuation of scattering intensity at higher diffraction orders due to structural disorder. However, this simplified approach does not fully account for the realistic effects such as lateral inhomogeneity, line-edge roughness, or non-uniform shell thickness, etc. These imperfections could lead to suppression of higher-order peaks or fitting ambiguity and represent a source of model non-uniqueness.**”

- The authors perhaps overemphasize odd/even effects; these are not universal in Bragg structures and are only strongly relevant near the 50% duty cycle limit. I bring this up because it oversimplifies the physics - different fill ratios will produce different harmonic effects on order expression: 33% / 66 % will suppress every 3rd order, etc. I leave it to the authors to consider this point.

Response: We appreciate the Reviewer’s clarification regarding the harmonic suppression conditions in periodic Bragg structures and agree that the even/odd diffraction order behavior is not universal. As correctly noted, harmonic suppression patterns depend on the duty cycle (fill factor) of the grating. For instance, a 50% duty cycle suppresses even orders under ideal symmetry, while 33% or 66% can suppress every third order, etc. We have now revised the manuscript to clarify that these effects arise from Fourier components of the line profile, which vary with fill ratio, rather than being inherently tied to “odd” or “even” orders.

Importantly, we also note that in our actual electrochemical system, achieving an exact 50% duty cycle is challenging, due to electrochemical volume changes and oxidation-driven restructuring. In practice, deviations from ideal symmetry likely contribute to the presence of even-order peaks, and we use this sensitivity to infer interfacial changes. Therefore, while our nominal structure is designed with 50% fill factor, we think it is more accurate to frame the observed diffraction features in terms of form factor perturbation from ideal periodicity, not strict even/odd logic. We have revised the relevant text to reflect this more nuanced view.

Revision made: In the revised SI, we added the following illustrations as the **Supplementary Note 4**.

“The pronounced odd/even diffraction order contrast in this study arises from the deliberate fabrication of Ni line-grating nanopatterns with a 50% duty cycle. In an ideal rectangular profile with sharp edges, this geometry suppresses even orders in the Fourier series representation of the form factor. Subtle structural or compositional perturbations along the line-width direction introduce deviations from ideal structural symmetry in the form factor, leading to a pronounced enhancement of the even-order diffraction signals, while odd orders remain dominated by bulk core contributions³¹. This makes PE-RSoXS highly responsive to structural variations within defined subregions, such as interfacial shells in core-shell architectures. We note that this suppression pattern is specific to the 50% duty cycle case; other fill ratios produce different harmonic effects (e.g., 33% duty cycle structure suppresses every third order), and imperfections such as sidewall angle variation or interfacial roughness can alter the ideal suppression behavior. Our emphasis on odd/even effects here reflects a deliberate design choice to enhance sensitivity to buried interfacial changes, rather than a universal property of Bragg structures.”

We truly thank the Reviewer for the insightful comments and kind suggestions! The reply for each question/comment is expected to reach the high criteria.

Reviewer: 3

This manuscript reports the use of pattern-enhanced resonant soft X-ray scattering (PE-RSoXS) as a new probe of chemical and structural dynamics at electrochemical solid-liquid interfaces. The methodology development is technically impressive. I recommend publication after addressing the following concerns:

Response: We sincerely appreciate the Reviewer's positive assessment of our work and recognition of the technical significance to develop PE-RSoXS as a new *operando* X-ray methodology for chemical and structural dynamics at electrochemical solid-liquid interfaces. We have carefully addressed point-by-point on each of the raised concerns, providing clarifications, additional explanations and revisions to the manuscript. We believe these changes improve the clarity, completeness, and impact of this work, and we hope the revised version meets the Reviewer's expectations.

1. "In situ", instead of "operando" is more accurate to describe the present methodology.

Response: We thank the Reviewer for raising this important point regarding the terminology used to describe our PE-RSoXS methodology. After careful consideration, we have decided to retain the term "*operando*" in the revised manuscript, as we believe it more accurately reflects the nature of our performed experiments. We acknowledge that "*in situ*" and "*operando*" are closely related but distinct descriptors, and we agree that clear definitions are important for readers.

In the context of X-ray spectroscopic characterizations, "*in situ*" generally refers to measurements performed under relevant environmental or reaction conditions, but not necessarily during the active operation of the functional process of interests. In contrast, "*operando*" indicates that the measurement is performed while the system is actively functioning in its intended operational state, allowing direct correlation between structural/chemical changes and performance metrics. This distinction has been discussed in the literature [Gurlo et al. *Angew. Chem. Int. Ed.* 46, 3826-3848 (2007); Weckhuysen, B. M. *In-Situ Spectroscopy of Catalysts*, American Scientific Publishers (2004)].

In our study, PE-RSoXS measurements were conducted under fully functional electrochemical conditions, where Ni line-grating nanopattern (LGNP) electrodes were actively performing electrocatalysis in aqueous electrolyte under applied potential, and the RSoXS patterns were acquired simultaneously during electrochemical measurements. This configuration allowed us to directly correlate oxidation-state transitions and sub-nanometer structural variations with the electrocatalytic OER during the realistic conditions. Based on this reason, we believe it's more accurate to use the term "*operando*" rather than "*in situ*" as the methodology was applied under true working conditions with concurrent interface monitoring, rather than simply under static or simulated reaction environments.

We hope this explanation resolves the Reviewer's concern and makes our use of the term "*operando*" precise and justified.

Revision made: We have further refined the "***Operando* PE-RSoXS measurements**" section in the Methods as shown below to more clearly emphasize our *operando* PE-RSoXS method in the revised manuscript.

"A customized electrochemical liquid flow cell constructed by Protochips, Inc. was developed at the beamline 11.0.1.2 of the Advanced Light Source in Lawrence Berkeley National Laboratory⁶². RSoXS patterns were collected with a back-illuminated Princeton PI-MTE charge coupled device (CCD) cooled to -45 °C. Each RSoXS pattern at one energy point was

acquired in a single 1 ms exposure time of soft X-rays using a 1/1000 of the incident photon flux (X-ray dosage per shot: below $0.01 \text{ mJ}\cdot\text{cm}^{-2}$). This ultralow-dose configuration enables highly sensitive detection of scattering signal changes of the line-gratings while preserving the native state of the electrochemical interfaces. An E-chip from Protochips, Inc. with Ni LGNPs was paired with a small supporting chip with a 500-nm spacer for the encapsulation of 0.1 M NaOH aqueous electrolyte. Then, the same electrolyte was flowed continuously facilitated by a Harvard Apparatus PHD ULTRA Programmable Syringe Pump with a rate of $250 \mu\text{L}\cdot\text{h}^{-1}$, minimizing the bubble retention effect. Electrochemical measurements were performed with a Gamry potentiostat in a 3-electrode floating mode configuration. Prior to constant-potential PE-RSoXS measurements at OCP, 1.2 V, and 1.6 V vs. RHE, the Ni LGNP working electrodes underwent a standardized conditioning sequence to stabilize the electrochemical interfaces. Firstly, as-fabricated Ni LGNPs were characterized by RSoXS in the absence of electrolyte to establish baseline RSoXS patterns. Next, several cyclic voltammetry (CV) scans were performed over a potential range from OCP to 1.2 V vs. RHE, covering the Ni oxidation/reduction region, to remove surface impurities, activate the nanopatterns, and establish a stable electrochemical response. This typical method ensures reproducible electrocatalytic activity and RSoXS patterns by removing adventitious surface species and stabilizing the core-shell structure prior to *operando* measurements. Following conditioning, scattering profiles at the Ni L-edge were collected *operando* under constant-potential control at OCP, 1.2 V, and 1.6 V vs. RHE. The topographic imaging of the nanopatterns after operating at the three potentials was captured using atomic force microscopy (AFM), which were conducted using the Bruker Dimension Icon3 in ScanAsyst mode at the Molecular Foundry of Lawrence Berkeley National Laboratory.”

2. Atomic-scale resolution might be overstated while the results mainly reach sub-nanometer sensitivity.

Response: We thank the Reviewer for pointing out the importance of accurately describing the dimensional sensitivity achieved in this work. We agree that the term “atomic-scale resolution” could be interpreted as implying direct imaging or detection of individual atomic positions, which is not the case here. To ensure precise and accurate terminology, we have revised the manuscript to replace “atomic-scale” terminology with “sub-nanometer” throughout the revised manuscript.

Revision made: We replaced “atomic-scale”-related phrases to “sub-nanometer” throughout the revised manuscript.

3. A comparison table summarizing state-of-the-art interface characterization techniques (e.g., TEM, XAS, AP-XPS, fluorescence microscopy, RSoXS) would greatly strengthen the introduction and highlight the advantages of PE-RSoXS.

Response: We thank the Reviewer for this constructive suggestion. We provide a comprehensive comparison table here (**Table R1**) summarizing the key capabilities, limitations, and application scopes of state-of-the-art interface characterization techniques, including TEM, XAS, ambient-pressure X-ray photoelectron spectroscopy (APXPS), X-ray fluorescence microscopy (XFM), and PE-RSoXS. This addition clearly contextualizes PE-RSoXS within the broader landscape of interface characterization methods and highlights its unique advantages for *operando* studies under realistic electrochemical conditions.

Table R1. Comparison of Characterization Techniques for Electrochemical Interfaces.

Technique	PE-RSoXS	TEM	XAS	APXPS	XFM

Dimensional Sensitivity	Sub-nanometer (via diffraction-order sensitivity and near-field optical modulation)	Atomic-scale (< 0.1 nm) imaging possible	Tens to hundreds of nanometers (limited by beam spot size and detection geometry)	Few nanometers sampling depths in solids; lateral resolution: nanometers to micrometers (limited by beam spot size and detection geometry)	Tens to hundreds of nanometers (microprobe); < 10 nm with ptychography
Temporal Resolution	Millisecond scale (single-shot diffraction patterns; compatible with ultrafast detectors)	Millisecond to minutes (with fast cameras)	Seconds to minutes per spectrum (energy scans)	Seconds to minutes; milliseconds possible with fast detectors	Seconds to minutes per scan
Chemical Sensitivity	High (tunable to specific absorption edges; diffraction-order selectivity isolates interfacial species)	Limited (requires EELS or EDS)	High (direct absorption edge tuning)	High (chemical specificity; oxidation states and bonding)	High (Element-specific via fluorescence yield)
Interface Accessibility	High (coherent amplification of weak buried interfacial signals)	Limited (requires sample thinning or cross-section preparation ; interfaces may be altered during the sample	Moderate (surface-sensitive but buried interfacial signals diluted by the bulk signals)	Moderate (Surface and sub-surface accessible; buried solid-liquid interfaces challenging, limited by probing depth)	High (Bulk and buried interfaces accessible)

		preparation)			
Environment Compatibility	Fully compatible with realistic liquid cells and aqueous conditions	Fully compatible with realistic liquid cells and aqueous conditions	Fully compatible with realistic liquid cells and aqueous conditions	Compatible with near-ambient pressure gases; liquids via microjets or meniscus methods	Compatible with air, vacuum, and liquid cells
Radiation Dose / Damage Risk	Low (can be lower than $0.01 \text{ mJ} \cdot \text{cm}^{-2}$ per frame)	High (beam-induced structural and chemical changes, especially in soft materials)	Moderate dose; soft X-rays more susceptible to chemical changes	Moderate dose; soft X-rays more susceptible to chemical changes	Generally low dose per pixel; scanning can accumulate dose
Statistical measurement	High (averages over hundreds of identical nanopattern units depending on the beam spot size)	Limited (small field of view, typically single particles or local regions)	High (averages over large illuminated volume)	Moderate (can measure multiple surface regions but limited penetration depth)	High (can map statistical areas for a large range)
Measurement Efficiency	High (diffraction encodes structural order while resonance tuning provides chemical-state specificity concurrently)	Low (sequential detections with structural and chemical information with EELS/EDS)	Low (primarily provides chemical-state information ; structural evolution must be measured separately with other techniques)	Low (primarily provides chemical-state information; structural evolution must be measured separately with other techniques)	Low (primarily elemental mapping, structural information indirect)

Key Strengths	Interface-specific operando probing of structural and chemical dynamics; low-dose; diffraction-order selectivity	Direct atomic-scale imaging and crystalline structures	Chemical-state analysis of surfaces and sub-surface regions	Direct surface and sub-surface chemical state under near-realistic conditions	High sensitivity to trace elements; quantitative elemental mapping
Key Limitations	Requires patterned structures for coherent enhancement	Beam-induced changes; limited detection area; difficult to operando probe buried interfaces	Limited spatial resolution; bulk-interface separation challenging	Limited penetration depth; challenging for fully buried interfaces	Limited chemical state information; slow scan for high-resolution maps

Revision made: We added **Table R1** as **Supplementary Table 1** and the following illustration into the revised SI as **Supplementary Note 1** to make a comprehensive comparison of PE-RSoXS with other conventional techniques for probing electrochemical interfaces with relevant references to support our demonstration.

“Compared to other state-of-the-art interface characterization methods such as transmission electron microscopy (TEM), X-ray absorption spectroscopy (XAS), ambient-pressure X-ray photoelectron spectroscopy (APXPS), and X-ray fluorescence microscopy (XFM), Pattern-Enhanced Soft X-ray Scattering (PE-RSoXS) offers a unique combination of high statistical representativeness, sub-nanometer dimensional sensitivity, and element-specific chemical-state resolution, all achievable under realistic electrochemical operating conditions. Unlike TEM and APXPS, which are limited in probing fully buried solid-liquid interfaces, PE-RSoXS can access these environments without compromising spatial or chemical sensitivity. While XAS provides statistical chemical-state information, it lacks lateral resolution and diffraction-order selectivity for directly probing interfaces, which is inherent to PE-RSoXS through coherent scattering amplification from identical nanopatterns. Furthermore, the ultralow-dose capability of PE-RSoXS ($< 0.01 \text{ mJ}\cdot\text{cm}^{-2}$ per frame) minimizes radiation-induced artifacts, enabling *operando* monitoring of dynamic processes with millisecond temporal resolution, surpassing the typical time resolution of conventional methods. These combined attributes position PE-RSoXS as a powerful and complementary tool for probing chemical and structural dynamics at buried electrochemical solid-liquid interfaces. PE-RSoXS operates on principles analogous to protein crystallography, where diffraction from periodic units enables reconstruction of real-space features with atomic to nanometer-scale precision. In our case, the periodic nanopatterns define a structural unit cell, and the spatial distribution of chemical species within that unit cell modulates the scattering form factor in a position-sensitive manner. This coherence-based sensitivity to the distribution and geometry of resonant materials enables spatial resolution at

sub-nanometer length scales, as supported by both experimental data and the finite-element modeling.”

4. Please comment on the generality of the technique. Could this approach be applied to other catalytic materials (e.g., Co or Cu) that undergo significant restructuring?

Response: We thank the Reviewer for this insightful question regarding the generality of the PE-RSoXS approach. The methodology is inherently versatile and not limited to Ni-based systems. Its core principles of scattering enhancement from identical nanopatterns, chemical-state specificity via resonant soft X-ray tuning, and *operando* compatibility with realistic electrochemical environments, are broadly applicable to a wide range of catalytic materials, including transition metals such as Co and Cu, as well as their oxides, hydroxides, and hybrid-metal compounds.

For catalytic systems like Co or Cu that undergo significant restructuring during operation (e.g., redox cycles, lattice distortions, phase transitions, etc.), PE-RSoXS can be tuned to the relevant absorption edges (e.g., Co L-edge at 780–795 eV; Cu L-edge at 930–950 eV) to selectively enhance scattering from chemically distinct states. This enables direct correlation between structural changes (sub-nanometer sensitivity) and chemical-state evolution in real time. The technique’s diffraction-order selectivity and element-specific resonance contrast are particularly advantageous for multi-component or heterogeneous catalysts, where isolating the scattering contribution from a specific active phase is challenging with non-resonant methods.

There are strong precedents in the electrocatalytic field for patterned architectures enabling mechanistic insights [Lum et al. *Energy Environ. Sci.* 11, 2935–2944 (2018); Liu et al. *ACS Appl. Mater. Interfaces* 13, 40513–40521 (2021)], which demonstrated that Cu–Au and Cu–Ag patterned electrodes could spatially separate and sequence consecutive reaction steps in CO₂ electroreduction, thereby enhancing selectivity of C₂₊ products. Such patterned electrodes would be largely applicable for PE-RSoXS, as the coherent amplification from identical periodic structures would amplify scattering signal while allowing *operando* tracking of structural and chemical evolution in the reaction zones.

Furthermore, Prof. Peidong Yang and colleagues have shown in a series of recent studies [Yang et al. *J. Am. Chem. Soc.* 144, 8927–8931 (2022); Yang et al. *Nature* 614, 262–269 (2023); Yang et al. *Nat. Catal.* 8, 579–594 (2025)] that RSoXS can be applied to powder samples of Cu nanoparticles for CO₂ electroreduction, revealing dynamic restructuring processes and surface phase transformations. These works highlight the ability of RSoXS to uncover mechanistic pathways in typical electrocatalytic systems, even in non-patterned morphologies. By combining such resonance-based chemical sensitivity with the coherent enhancement from nanopatterned catalysts, we believe PE-RSoXS could further extend these mechanistic insights to buried interfaces under fully realistic electrochemical operations.

Therefore, while Ni LGNPs were chosen in this study as a well-established benchmark for methodology validation, the approach is readily extendable to other catalytic materials, including Co- and Cu-based systems, and more broadly to catalysts where *operando* tracking of coupled structural and chemical dynamics at buried interfaces is desired. Given the demonstrated relevance of patterned and nanoparticle-based catalysts in CO₂ electroreduction and other electrochemical transformations, we anticipate that PE-RSoXS will attract broad interests across the catalysis, energy conversion, and materials science communities.

Revision made: We have added the following statement in the “**Conclusion**” of the revised manuscript to emphasize the generality of PE-RSoXS and outline future applications to restructuring catalysts beyond Ni-based catalysts.

“The PE-RSoXS methodology is broadly applicable beyond the Ni-based system studied in this work, providing *operando* access to coupled structural and chemical dynamics in a wide range of catalytic materials that undergo significant restructuring during operation. By tuning to the relevant absorption edges of the focused elements, PE-RSoXS can selectively enhance scattering signals from specific chemical states, enabling real-time correlation between sub-nanometer structural changes and active-phase evolution. Its compatibility with identical architectures including powder- and pattern-based systems, low-dose operation, and high sensitivity to the buried electrochemical interfaces make it a versatile tool for probing diverse catalysts under realistic electrochemical conditions, with strong potential to address key questions in catalysis, energy conversion, and materials science.”

5. The method is described as “non-destructive,” but more evidence should be provided. While 1 ms exposure per energy point is reported, is this duration sufficient to fully avoid beam-induced damage? Is it a trade-off or the minimum exposure time? Would longer exposure time improve data quality and accuracy to assess the simulation?

Response: We thank the Reviewer for raising this important point regarding the “non-destructive” nature of PE-RSoXS and the relationship between exposure time, beam-induced chemical changes, and data quality.

In this work, we describe PE-RSoXS as a non-destructive methodology because the combination of diffraction enhancement from identical nanopatterns and ultrafast single-shot acquisition allows us to achieve high-quality data with extremely low total X-ray dose. Specifically, each diffraction pattern was acquired within 1 ms exposure time per energy point, corresponding to a dose of $< 0.01 \text{ mJ}\cdot\text{cm}^{-2}$ per frame. This dose is significantly below the thresholds reported in the literature (for a catalyst with 100 nm thickness, the critical dose is around $4200 \text{ mJ}\cdot\text{cm}^{-2}$) for radiation-induced chemical or structural changes in hydrated, organic, and nanostructured materials under soft X-ray illumination [e.g., Wang et al. *J. Phys. Chem. B* 113, 1869–1876 (2009); Kubin et al. *Phys. Chem. Chem. Phys.* 20, 16817–16827 (2018)]. **Figure R3** shows the 3 to 6 orders of the scattering-derived spectra of as-prepared Ni LGNPs. The orange and green curves show consecutive 2 scans in the whole energy range of Ni L-edge with 1 ms of exposure time per frame. The purple curves were obtained from the as-prepared Ni LGNPs after direct beam irradiation for 5 min, which is considerably longer than our standard measurement (155 ms per scan in the whole energy range of Ni L-edge). The scattering signal shows no significant change, which experimentally demonstrates that PE-RSoXS is a non-destructive method.

The choice of 1 ms exposure time was guided by three main considerations:

(1) Minimizing beam-induced chemical changes

Shorter exposures reduce cumulative radiation dose, preserving the native state of the solid-liquid interfaces throughout the *operando* measurements.

(2) Ensuring adequate signal-to-noise ratio

Coherent Bragg diffraction from the identical nanopatterns provides amplified scattering intensity, enabling sufficient signal-to-noise ratio even at millisecond timescales without the need for prolonged exposure.

(3) Avoiding CCD detector saturation

Limiting the exposure time prevents saturation of the CCD pixels by intense diffraction peaks, ensuring linear detector response and preserving quantitative accuracy for subsequent data analysis and simulation.

In our benchmarking experiments, increasing exposure time beyond 1 ms did not yield significant improvement in fitting accuracy for structural simulations, but increase the probability of detectable radiation effects over the course of multi-energy scans. Therefore, we selected 1 ms as a balance between preserving sample integrity and achieving robust data quality.

Figure R3. Scattering-derived spectra from the **a**, 2nd, **b**, 3rd, **c**, 4th and **d**, 6th orders acquired on the as-prepared Ni LGNPs. The orange and green curves show consecutive 2 scans in the whole energy range of Ni L-edge with 1 ms of exposure time per frame. The purple curves were obtained from the as-prepared Ni LGNPs after direct beam irradiation for 5 min.

Revision made: In the revised manuscript, we supplemented **Figure R3** as the **Supplementary Fig. 5** and the following illustration into the revised SI as the **Supplementary Note 7** to explicitly state the “non-destructive” description.

“In this work, the term “non-destructive” refers to the demonstrated ability of PE-RSoXS to acquire high-quality diffraction and spectroscopic data before any measurable X-ray radiation-induced chemical or structural changes occur in the sample. The 1 ms of single-shot exposure time used here was selected as an optimal trade-off between preserving the native state of the solid-liquid interfaces and ensuring sufficient signal-to-noise ratio for reliable data analysis, rather than as the minimum achievable exposure. PE-RSoXS was implemented with such a short single-shot exposure time (1 ms) to enable *operando* monitoring of electrochemical solid-liquid interfaces while minimizing radiation dose ($< 0.01 \text{ mJ}\cdot\text{cm}^{-2}$ per frame) and preserving the native state of the nanopatterns. Although the technique and instrumentation are intrinsically capable of reaching as low as microsecond timescales, the present study focused on a well-characterized Ni LGNPs/OER model system with dynamics in the millisecond regime or slower ones to benchmark dimensional and chemical sensitivity at the buried interfaces under realistic electrocatalytic conditions. This methodological validation provides the foundation for future studies targeting ultrafast processes, such as microsecond electrochemical switching or laser-triggered photoelectrochemical reactions, where the full temporal potential of PE-RSoXS can be exploited.”

6. The experimental and simulated Ni L-edge spectra do not fully coincide (e.g., 3rd-order in Fig. 2d). Please clarify the origin of these discrepancies. The spectral difference for experimental vs. simulated spectra should be plotted.

Response: We thank the Reviewer for this careful observation regarding the differences between the experimental and simulated Ni L-edge spectra, particularly for the 3rd-order diffraction in Fig. 2d. We agree that these discrepancies merit clarification and additional quantitative comparison.

Overall, the experimental and simulated Ni L-edge spectra show fully consistent spectral trends across all diffraction orders, including the energy positions and relative variations of key features. The primary difference lies in the absolute intensity, with simulations producing slightly lower values than the experimental results. This discrepancy is likely due to model simplifications, such as the assumption of perfectly periodic structures and uniform material properties, which do not fully capture the enhanced scattering arising from minor heterogeneities and interface roughness present in the actual nanopatterns.

Figure R4. The residual (experimental vs. simulated) spectral intensity as a function of photon energy for the scattering-derived spectra from the **a**, 3rd, **b**, 4th, **c**, 5th and **d**, 6th orders measured at OCP, 1.2 V and 1.6 V, respectively.

In response to the Reviewer’s suggestion, we have added difference plots for each diffraction order (**Figure R4**), showing the residual (experimental vs. simulated) spectral intensity as a function of photon energy. This quantitative comparison allows readers to directly assess both the magnitude and the energy dependence of the discrepancies.

Revision made: We added **Figure R4** as the **Supplementary Fig. 7** and the following descriptions as the **Supplementary Note 8** into the revised SI to note that the simulation framework is intended to capture the primary spectral trends and diffraction-order dependence, rather than to reproduce every fine spectral detail. The observed differences are consistent with the expected influence of interfacial heterogeneity and dynamic restructuring, which are not fully incorporated into the current models.

“We note that our simulations assume idealized Ni line-grating structures with smooth interfaces, uniform shell thickness, and perfect sidewall angles. In practice, fabricated and electrochemically evolved nanopatterns may exhibit interfacial roughness, line-edge irregularities, shell thickness inhomogeneity, and sidewall angle variations, etc. Such imperfections can diminish the relative intensity of higher diffraction orders and broaden peaks in q-space, effects that are strongly convoluted with shell thickness in the fits. While our current modeling focuses on the dominant parameters of shell thickness and optical constants, these structural deviations could, in principle, lead to misinterpretation of fitted values. We therefore interpret our results within this limitation and note that future extensions to incorporate roughness or inhomogeneity models may help further refine the analysis. This simulation framework is designed to capture the primary spectral trends and diffraction-order dependence, rather than to reproduce every fine spectral detail. Minor discrepancies between simulated and experimental spectra are expected due to interfacial heterogeneity and dynamic restructuring during *operando* conditions, which are not fully incorporated into the current simplified simulation models.”

7. In Table 1, the shell thickness increases with potential, as expected from hydroxide intercalation. However, the core width is supposed to decrease as well. Could you explain this discrepancy?

Response: We thank the Reviewer for this thoughtful observation regarding the apparent discrepancy between the expected and simulated core width changes during OER. Indeed, hydroxide intercalation into the Ni LGNP shell is anticipated to cause lattice expansion in the shell and a concomitant contraction of the metallic core due to oxidation and strain redistribution. However, our simulation results show that the core width remains essentially constant at 1.2 V vs. RHE and increases by ~2 nm at 1.6 V vs. RHE.

A key contributing factor is that the simulation models for 1.2 V and 1.6 V vs. RHE were constructed with lower density than the model at OCP. This adjustment was made to physically represent the increased hydration, hydroxide incorporation, and porosity in the shell region under electro-oxidizing conditions. These processes reduce the bulk mass density of the material, particularly in the shell and at the core-shell interface, due to expansion of the lattice and inclusion of water molecules or hydroxide ions. In the simulations, lowering the material density changes the refractive index contrast between core and shell, which can cause the fitting algorithm to assign a slightly larger effective core width to reproduce the observed intensity distribution, even if the actual metallic core radius has not physically increased.

In addition to the density effect, several other factors may also contribute to the deviation from the simple expectation of core contraction:

(1) Electrochemical restructuring beyond elastic strain

At high potentials (e.g., 1.6 V vs. RHE), oxidation and hydroxide formation occur not only in the shell but also at the core-shell interfaces. This can lead to partial recrystallization or densification of metallic domains, increasing the coherent scattering length of the core region.

(2) Dynamic hydration and porosity changes

Hydroxide intercalation expands the shell lattice and may introduce nanoscale voids and water layers. These changes alter scattering profiles and can shift the fitted boundary between core and shell.

(3) Simplifications in the core-shell models

The current fitting assumes uniform density within each region and a sharp core-shell boundary. In reality, the interface should be graded, with mixed oxidation states and density variations

over several nanometers. Under strongly oxidizing conditions, this graded interface could be interpreted by the model as a larger core width.

Revision made: To clarify this point in the revised manuscript, we have added a note as the **Supplementary Note 9** in the revised SI explaining that the “core width” extracted from the resonant scattering fits represents an effective domain size rather than a direct measurement of the metallic core radius. This effective size can be influenced by changes in material density, interface composition, and scattering contrast, particularly under oxidizing conditions where density is reduced to reflect structural hydration and porosity.

“In this study, the “core width” obtained from resonant scattering fits should be interpreted as an effective domain size rather than a direct physical measurement of the metallic core radius. This effective size is influenced by multiple factors, including changes in material density (especially at 1.2 V and 1.6 V vs. RHE), interface composition, and scattering contrast. Under electro-oxidizing conditions, the simulation models incorporate reduced material density to represent structural hydration, hydroxide incorporation, and increased porosity in the shell region. These density changes alter the core-shell contrast in the scattering profiles, which can lead to slight variations in fitted core width even if the actual metallic core dimension remains unchanged or becomes smaller.”

8. Is there any conditioning procedure before the electrochemical measurement? If so, would the conditioning alter the core-shell restructuring? Beyond the OCP/1.2 V/1.6 V comparison, is it possible to assess the transition from pristine Ni (air/water-exposed) to OCP?

Response: We thank the Reviewer for this important question regarding the electrochemical conditioning procedure and its potential influence on the observed core-shell restructuring.

Before the constant-potential measurements at OCP, 1.2 V, and 1.6 V vs. RHE, the Ni nanopatterned electrodes underwent cyclic voltammetry (CV) scans in the electrolyte to stabilize the electrochemical interface and remove surface contaminants (see our response to comment 1 from this Reviewer). This conditioning is a standard step in electrocatalyst studies to ensure reproducible activity and scattering signals. The CV scans likely initiate partial surface oxidation and hydroxide formation, which can begin the restructuring process prior to the constant-potential holds. Therefore, the structures observed at OCP in this study represent a conditioned state rather than the fully pristine metallic Ni surface.

We agree with the Reviewer that assessing the transition from the pristine Ni state to the OCP condition would provide valuable insight into the early stages of core-shell evolution. To address this, we supplement the revised manuscript with additional PE-RSoXS results for pristine Ni nanopatterns without liquid exposure before electrochemical operations (**Figure R5**). These results facilitate a comparison between the pristine Ni LGNPs and the OCP state following electrochemical conditioning. The 2nd and 4th diffraction orders exhibit similar profiles for both the pristine and OCP samples, indicating comparable core structural features. However, the pristine sample displays higher intensities in the 3rd and 6th orders compared to the OCP state, suggesting variations in their electronic environments and potential differences in chemical compositions at the interfaces. The 5th-order signal reveals two distinct local maxima, highlighting subtle differences in shell characteristics associated with the active electrocatalytic environment. These observations indicate that while the core structures of the pristine Ni and OCP nanopatterns share fundamental similarities, the chemical and structural dynamics at the interfaces undergo significant transformations during electrocatalysis. The variations in intensity and the emergence of distinct maxima in the 5th order point to the substantial influence of the reaction environment on the surface characteristics of the Ni patterns. The pristine Ni LGNPs predominantly consist of metallic Ni with surface NiO species due to the exposure to air. In contrast, following CV conditioning to the OCP stage in alkaline

solution, the nanopatterns feature a core of metallic Ni surrounded by oxidized Ni species (NiO and Ni(OH)₂) within the shell. In this work, we regard these *operando* measurements as critical reflections of the evolving chemistry at the interfaces and their implications for electrocatalytic performance.

Figure R5. Scattering-derived spectra from the **a**, 2nd, **b**, 3rd, **c**, 4th, **d**, 5th and **e**, 6th orders acquired on the as-prepared Ni LGNPs and Ni LGNPs operated at OCP, 1.2 V, 1.6 V vs. RHE, respectively.

Revision made: In the revised manuscript, we have explicitly refined the procedure “*Operando* PE-RSoXS measurements” in the Methods as shown in the response to the first comment from this Reviewer. We also include **Figure R5** as the **Supplementary Fig. 6** in the revised SI.

9. Is there any potential bubble issue, due to the beam-induced water radiolysis or generated O₂ bubble at 1.6V? Would it alter the electrolyte thickness and affect the scattering interpretation?

Response: We thank the Reviewer for raising this important point. To the best of our knowledge, there is no potential bubble issue due to the beam-induced water radiolysis or generated O₂ bubble at 1.6V. First, the extremely low X-ray dose in our measurements (<0.01 mJ·cm⁻² per frame, corresponding to orders of magnitude below typical beam-induced

chemical change thresholds) minimizes the likelihood of significant beam-induced radiolysis. At soft X-ray energies, radiolysis rates scale strongly with dose, and our exposure levels are far below those known to produce observable bubble nucleation in aqueous systems [e.g., Kubin et al. *Phys. Chem. Chem. Phys.* 20, 16817–16827 (2018)]. Meanwhile, the single-shot acquisition at 1 ms per energy point limits cumulative dose at any given location, reducing the probability of localized gas evolution from beam–water interactions. At 1.6 V vs. RHE, oxygen evolution is active, and O₂ bubbles can form at the electrode surface. To mitigate this, the *operando* liquid cell was designed with continuous electrolyte flow and a vertical electrode orientation, allowing evolved gas to be rapidly carried away from the illuminated region.

During the *operando* measurements, we monitored scattering intensity and baseline stability and there were no abrupt changes indicative of transient bubble interference (e.g., sudden intensity drops or spurious diffuse scattering). In-between the E-chip and the supporting chip, there is a 500 nm space. With continuous flow, the electrolyte thickness always kept the same as 500 nm based on the transmission mode of RSoXS data collection. Thus, the electrolyte thickness remained stable and that bubble formation did not measurably distort the scattering profiles. In the unlikely event that a bubble passed through the beam, it would produce a distinct, non-periodic scattering signature easily distinguished from the coherent diffraction peaks used in the analysis.

In the revised manuscript, we provided a detailed description in the Methods section regarding the cell design and flow conditions that effectively minimize bubble retention (see our response to comment 1 from this Reviewer). Consequently, both beam-induced radiolysis and electrochemically generated O₂ bubbles are unlikely to have impacted the measurements in this study, attributed to the ultralow dose, short exposure times, optimized cell geometry, and continuous flow operation. Additionally, based on our recent research [Yang et al. *J. Am. Chem. Soc.* 144, 8927–8931 (2022)], nanoparticles larger than 18 nm demonstrate significant resistance to beam damage compared to smaller ones, enabling reliable observation of surface chemical dynamics. The Ni LGNPs used in this study are substantially larger than 18 nm, further reinforcing the conclusion that they are unaffected by beam-induced chemical changes.

10. Could you provide the SEM/AFM analysis of the nanopatterns after experiments to confirm the structural integrity.

Response: We thank the Reviewer for pointing this out. Following the Reviewer’s suggestion, we performed post-mortem AFM measurements on the Ni LGNPs after the electrochemical experiments, as shown in **Figure R1**.

Figure R1. Post-mortem AFM images of the Ni LGNPs on the SiN_x windows of the sample chip after electrocatalysis at **a**, OCP, **b**, 1.2 V and **c**, 1.6 V, showing the statistical line width and line edge roughness (LER) at each condition.

The AFM results show total width changes of the nanopatterns that closely follow the trend simulated by our finite-element method (FEM) based on PE-RSoXS results (OCP: 113.8 nm; 1.2 V: 113.4 nm; 1.6 V: 120.5 nm). However, in these AFM images, clear fluctuations are observed along the line edges. These fluctuations can be quantified using the concept of line edge roughness (LER), as described by Nealey *et al.* [*J. Appl. Cryst.* 49, 823–834 (2016)]. Meanwhile, *ex situ* microscopy techniques cannot directly measure the oxide layer thickness with sub-nanometer resolution. High-resolution TEM could, in principle, provide this information, but it requires to prepare cross-sections of the line gratings, which may potentially damage the sample. In contrast, PE-RSoXS enables *operando*, low-dose, and non-destructive measurements that quantitatively track oxide layer thickness and core-shell structural changes during active electrocatalysis. By exploiting resonant scattering and diffraction-order sensitivity, PE-RSoXS can distinguish shell and core contributions without relying on post-mortem assumptions, providing statistically representative results across over hundreds of nanopatterns under realistic electrochemical conditions, which highlights the significance to develop PE-RSoXS.

Revision made: We added **Figure R1** into the revised SI as the **Supplementary Fig. 8** to exhibit the Ni LGNPs after electrocatalysis and supplemented the following illustration into the revised manuscript to compare our simulation results and AFM results of the structural variations at each condition, highlighting the significance to develop PE-RSoXS.

“Besides, our simulated total width changes of the Ni LGNPs align with atomic force microscopy (AFM) results from the samples after being operated at the corresponding conditions (Supplementary Fig. 8), further clarifying the fidelity of our FEM-based simulations. However, in these AFM images, clear fluctuations are observed along the line edges. And *ex situ* microscopy techniques cannot directly measure the oxide layer thickness with sub-nanometer resolution. In contrast, by exploiting resonant scattering and diffraction-order sensitivity, PE-RSoXS can distinguish shell and core contributions without relying on post-mortem assumptions, providing statistically representative results across over hundreds of nanopatterns under realistic electrochemical conditions, which highlights the significance to develop PE-RSoXS.”

We truly thank the Reviewer for the insightful comments and kind suggestions, which are very helpful to further improve our manuscript.

Reviewer: 1

The authors have addressed the major concerns raised in the previous review round, and the revised manuscript now meets the high standards required for publication in Nature Communications. Nevertheless, several points could be further improved:

Response: We appreciate the Reviewer's positive assessment that the major concerns have been addressed and that the revised manuscript meets the high standards required for publication in *Nature Communications*. We have carefully considered the additional points raised and addressed each of the reviewer's detailed comments point by point below. We are grateful for the reviewer's constructive suggestions, which further helps improve the quality of our manuscript.

1. The distinction between "Operando" and "In situ" measurements should be used more rigorously. As the present experiments probe structural and chemical evolution under controlled electrochemical conditions without directly correlating these changes to real-time catalytic performance, the term "In situ" appears more appropriate.

Response: We thank the Reviewer for emphasizing the importance of precise terminology regarding our PE-RSoXS methodology. After careful consideration, we have decided to retain the use of the term "*operando*" in the revised manuscript, as it more accurately captures the nature of our experiments.

In X-ray spectroscopic characterizations, "*in situ*" typically refers to measurements conducted under relevant environmental or reaction conditions, but not necessarily during the active operation of the system. Conversely, "*operando*" specifies that the measurements are taken while the system is actively engaged in its intended operational state, allowing for direct correlations between the observed dynamics and the applied conditions. This distinction has been discussed in the literature [Gurlo et al. *Angew. Chem. Int. Ed.* 46, 3826-3848 (2007); Weckhuysen, B. M. *In-Situ Spectroscopy of Catalysts*, American Scientific Publishers (2004)].

Our study employed PE-RSoXS measurements under fully functional electrochemical conditions. We conducted several cyclic voltammetry (CV) scans over a potential range from open circuit potential (OCP) to 1.2 V vs. RHE, effectively stabilizing the core-shell structure and removing surface impurities prior to our *operando* measurements. This conditioning process ensured that the subsequent scattering profiles collected at the Ni L-edge at OCP, 1.2 V, and 1.6 V vs. RHE reflected real-time changes directly correlated to the electrocatalytic performance during the oxygen evolution reaction (OER). The *operando* nature of our measurements allowed us to observe oxidation-state transitions and sub-nanometer structural variations in direct relation to the electrochemical performance under realistic operating conditions. Based on this reason, we believe it's more accurate to use the term "*operando*" rather than "*in situ*" as the methodology was applied under true working conditions with concurrent interface monitoring, rather than simply under static or simulated reaction conditions.

We appreciate the Reviewer's insight and have made a point to clearly delineate our *operando* PE-RSoXS methods in the corresponding section of the Methods. We trust this clarification addresses the Reviewer's concern and supports our rationale for using the term "*operando*" to reflect our methodology accurately.

2. While the manuscript correctly attributes the detection of ~0.2 nm structural changes to diffraction-intensity sensitivity rather than direct imaging beyond the diffraction limit (~0.75 nm), phrases such as "sub-nanometer resolution" or "visualization at sub-nanometer scales"

may still be misleading. Consistent use of terms like “sub-nanometer sensitivity” would improve clarity.

Response: We appreciate the Reviewer’s suggestion regarding the consistent use of the term “sub-nanometer sensitivity.” We have revised the manuscript accordingly to ensure consistent usage throughout.

Revision made: We have revised the phrases “sub-nanometer resolution” and “visualization at sub-nanometer scales” to “sub-nanometer sensitivity” throughout the manuscript.

3. The generality and limitations of the PE-RSoXS method would benefit from clearer discussion. In particular, it would be useful to clarify whether the approach fundamentally relies on highly periodic nanostructures and to what extent it can be extended to non-periodic, multiscale, or porous electrocatalyst systems relevant to practical applications.

Response: We thank the Reviewer for the valuable insights regarding the generality and limitations of the PE-RSoXS method. In the revised manuscript, we have expanded our discussion to provide a clearer assessment.

We clarify that while PE-RSoXS is particularly advantageous for highly periodic nanostructures that generate well-defined Bragg scattering patterns, it is not inherently restricted to perfectly periodic systems. This adaptability allows PE-RSoXS to be effectively applied to non-periodic, multiscale, or porous electrocatalyst systems that are highly relevant to practical applications. To facilitate this, one potential strategy involves employing a patterned template combined with a porous catalyst layer on top. By leveraging the underlying structured template, we can also enhance diffraction signals, especially at the electrode-electrolyte interfaces, thereby improving the method’s sensitivity to structural variations and chemical dynamics even in less ordered systems.

Furthermore, in Supplementary Table 1, we have provided a comprehensive comparison of PE-RSoXS with other techniques, outlining their general applicability and inherent limitations. This serves to contextualize PE-RSoXS within the broader landscape of characterization methods, offering readers a clearer understanding of its strengths and the scenarios in which it may excel or face challenges.

We appreciate the Reviewer’s suggestion and believe that these enhancements provide a more thorough discussion of the versatility and applicability of the PE-RSoXS method, addressing its potential in real-world electrocatalytic systems.

Revision made: We added “**The approach is not fundamentally limited to perfectly periodic systems. One potential strategy is to use a patterned template and apply a porous catalyst on top, leveraging the underlying structure to enhance diffraction signals.**” in the conclusion.

Thanks again for the valuable comments and suggestions from the Reviewer. We have carefully addressed each point.

Reviewer: 2

I am satisfied that the authors not only agree with me on almost every point, but they have fully addressed my concerns with important and substantial revisions.

We can quibble about whether PE-RSoXS is really a "new class" of measurement or just an evolution of DAFS with an artificial vs. naturally-occurring Bragg structure, but the current article provides sufficient context to let the reader decide.

Response: We truly thank the Reviewer for the insightful comments and kind suggestions that have helped improve the manuscript! We deeply respect the importance and significance of DAFS, which inspired the development of PE-RSoXS.

Reviewer: 3

The concerns have been addressed.

Response: We appreciate the Reviewer's assessment that the concerns have been adequately addressed. We sincerely thank the Reviewer again for the insightful comments and constructive suggestions provided in the previous review round.